



# Effects of dimensionality on the performance of hydrodynamic models

Mayra Ishikawa[1], Wendy Gonzalez[2], Orides Golyjeswski[3], Gabriela Sales[3], J. Andreza Rigotti[4], Tobias Bleninger[5], Michael Mannich[5], Andreas Lorke[1]

[1]Institute for Environmental Sciences, University of Koblenz-Landau, Landau, 76829, Germany
[2]Institute for Water and River Basin Management, Karlsruhe Institute of Technology, Karlsruhe, 76131, Germany
[3]Graduate Program in Environmental Engineering, Federal University of Paraná, Curitiba, 82590-300, Brazil
[4]Postgraduate Program in Water Resources and Environmental Engineering, Federal University of Paraná, Curitiba, 81531-990, Brazil
[5]Department of Environmental Engineering, Federal University of Paraná, Curitiba, 82590-300, Brazil

*Correspondence to:* Mayra Ishikawa (ishikawa@uni-landau.de)

**Abstract.** Numerical models are an important tool for simulating temperature, hydrodynamics and water quality in lakes and reservoirs. Existing models differ in dimensionality by considering spatial variations of simulated parameters (e.g., flow velocity and water temperature) in one (1D), two (2D) or three (3D) spatial dimensions. The different approaches are based on different levels of simplification in the description of hydrodynamic processes and result in different demands in computational power. The
aim of this study is to compare three models with different dimensionality and to analyze differences between model results in relation to model simplifications. We analyze simulations of thermal stratification, flow velocity, and substance transport by density currents in a medium-sized drinking water reservoir in the subtropical zone, using three widely used open-source models: GLM (1D), CE-QUAL-W2 (2D) and Delft3D (3D). The models were operated with identical initial and boundary conditions over a one-year period. Their performance was assessed by comparing model results with measurements of temperature, flow velocity
and turbulence. Results show that all models were capable of simulating the seasonal changes in water temperature and stratification. Flow velocities, only available for the 2D and 3D approaches, were more challenging to reproduce, but 3D simulations showed closer agreement with observations. With increasing dimensionality, the quality of the simulations also increased in terms of error, correlation and variance. None of the models provided good agreement with observations in terms of mixed layer depth, which also affects the spreading of inflowing water as density currents, and the results of water quality models
that build on outputs of the hydrodynamic models.

## 1 Introduction

A large variety of different numerical models have been used for simulating temperature and hydrodynamics in lakes and reservoirs, as well as the biogeochemical and ecological processes that depend on it (e.g. Dissanayake et al., 2019; Guseva et al., 2020; Wang et al., 2020; Xu et al., 2021). While the mechanistic description of underlying physical processes is identical or at
least similar in all models, they differ in their dimensionality, i.e. the number of spatial dimensions considered in the model. One-dimensional (1D) models usually resolve the vertical direction only (water depth), while considering homogeneity of all relevant quantities along horizontal directions. They are attractive due to their easy connection to ecological and biogeochemical modules. In addition, the comparably low number of required input parameters and fast computational time allow evaluations of scenarios and sensitivity analyses, which facilitate their application for assessing long-term dynamics and resilience of lakes and
reservoirs in response to climatic, hydrological and land usage changes (Bruce et al., 2018; Sabrekov et al., 2017; Hipsey et al.,



2019). Therefore 1D models such as DYRESM (Imberger and Patterson, 1981; Hetherington et al., 2015), SimStrat (Goudsmit et al., 2002; Stepanenko et al., 2014), and GLM (Hipsey et al., 2019; Fenocchi et al., 2017; Bruce et al., 2018; Soares et al., 2019), have been extensively used in scientific and applied studies. On the other hand, detailed studies of hydrodynamic effects and spatially varying flow and transport mechanisms, such as density currents at river inflow locations, require models with a higher

dimensionality. Two-dimensional (2D) models can provide additional insights in the hydrodynamics of lakes and reservoirs while keeping computational costs low when compared to 3D models. The 2D models that neglect variations in the vertical dimension – 2DH, are suitable for shallow lakes, where gradients along depth are minor, but they are mainly used for flood maps, river flows, hydraulics structures, and sediment transport. Alternatively, the models resolve the vertical and one horizontal (longitudinal) dimension – 2DV, suitable for elongated deep-water bodies where vertical thermal stratification plays a major role, e.g. CE-QUAL-

W2 (Gelda et al., 2015; Kobler et al., 2018; Mi et al., 2020). Finally, three-dimensional (3D) models provide highly detailed spatial data, but require larger computational effort in terms of time and storage. Regardless their complexity, 3D models are widely applied, e.g. POM (Beletsky and Schwab, 2001; Song et al., 2004), ELCOM (Carpentier et al., 2017; Marti et al., 2011; Zhang et al., 2020) and Delft3D-FLOW (Soulignac et al., 2017; Bermúdez et al., 2018; Baracchini et al., 2020; Guénand et al., 2020).

The choice of model dimensionally often represents a trade-off between required accuracy, availability of boundary conditions and

computational costs. Different models can complement each other and model intercomparisons can significantly contribute to process-understanding of the studied system, as well to assessment of model limitations. Within the framework of the Lake Model Intercomparison Project (LakeMIP, Stepanenko et al., 2010), the performance of different 1D models was compared for a number of reference sites, targeting also at the improvement of model parameterizations (Stepanenko et al., 2013; Stepanenko et al., 2014; Thiery et al., 2014; Guseva et al., 2020). Perroud et al. (2009) compared four different 1D models, that were previously applied to

small water bodies, to the large Lake Geneva, and Mesman et al. (2020) investigated the performance of three 1D models under extreme weather events like storms and heat waves. Most of the comparison among 3D models focused on systems where circulation patterns and internal waves had a major influence. For example, Huang et al. (2010) compared three 3D models for Lake Ontario, where variations in surface temperature are caused by circulations patterns and upwelling / downwelling of the thermocline. Dissanayake et al. (2019) applied ELCOM and Delft3D to simulate internal wave motions and surface currents in

Upper Lake Constance. Zamani and Koch (2020) compared AEM3D and MIKE3 ED models in a reservoir with complex morphology.

Comparison of models with different dimensionality is more difficult, since the interpretation of each result also depends on model considerations and simplifications. Polli and Bleninger (2019) compared temperatures from the MTCR-1 (1D model) and Delft3D, and found that 1D modeling may provide similar information as 3D models in terms of thermal structure. Therefore, the study

recommended one-dimensional models as a first approach for assessing reservoir stratification patterns, and the application of a 3D model, if the horizontal substance transport is of interest. This information is relevant for simulations of lakes and reservoirs with sparse information on boundary conditions (Polli and Bleninger, 2019; Soares et al., 2019). Following the same idea, Man et al. (2021) recommended the application of a 1D model for parameter estimation that are subsequently used in 3D simulations, because of the shorter computational time of the former. In their simulations the 1D model had good results only for specific

periods (when stratification or mixing were stable). The Geologic Survey of Israel and Tahal (Gavrieli et al., 2011) developed numerical models with three different dimensionalities for the simulation of the hydrodynamics and temperature stratification of the Dead Sea: a 1D model using the software 1D-DS-POM, a 2D laterally-averaged model using CE-QUAL-W2 and a 3D model using the software POM2K. The models were used in a complementary manner, taking advantage of the strengths of each one:





The 1D model was used to simulate decades, in order to study future scenarios, the 2DV model was used to investigate the changes in the thermal structure of the reservoir due to changes in the multiple inflows and the 3D model allowed the study of currents and the 3D thermohaline structure. Nevertheless, the performance of the three different model approaches was not compared within this study, neither comparisons with respect to velocities where shown. It is important to remark at this point that the selection of a higher dimensionality does not imply better simulations results (Wells, 2020). DeGasperi (2013) compared the performance of CE-QUAL-W2 and CH3D-Z (3D model) in simulating the water temperature of Lake Sammamish in the USA. Both models presented similar results with slightly better performance statistics for the 2DV model. Al-Zubaidi and Wells (2018) evaluated the capacity of CE-QUAL-W2 and a three dimensional adapatation of the same software known as W3 in modeling the temperature stratification at Laurance Lake, Oregon, USA. For this study, the predictions of both models were in comparable agreement with measurements, but to run the 3D model was 60 times more expensive in terms of computational time.

The selection of a model in terms of its dimensionality is closely related to the objectives of the study, the water body characteristics, and the targeted computational costs. The aim of this study is to compare three models with different dimensionality and to analyze the results based on the simplification of the physical processes caused by model dimensionality. We analyze simulations of thermal stratification, horizontal flow velocity, and substance transport by density currents in a medium-sized drinking water reservoir in the subtropical zone using three widely used open-source models: GLM (1D), CE-QUAL-W2 (2D) and Delft3D-FLOW (3D). The models were run with identical initial and boundary conditions over a one-year period. Their performance was assessed by comparing model results with measurements of temperature, flow velocity and turbulence over a one-year period. We aim at providing a reference study supporting the selection of models and the assessment of model accuracy, as well as at improving the mechanistic understanding of model performance at reduced dimensionality.

## 2 Description of the models

### 2.1 General Lake Model (GLM)

General Lake Model (version 3.1.8) (Hipsey et al., 2019) is a one-dimensional vertical model, freely available, designed to simulate the water balance and the vertical stratification of lacustrine systems. The model computes the vertical profiles of temperature, salinity and density by considering hydrological and meteorological forcing. GLM adopts a flexible Lagrangian layer structure (Imberger et al., 1978; Imberger and Patterson, 1981), which allows the layer thicknesses to change dynamically by contraction and expansion, according to density changes driven by surface heating, mixing, inflows and outflows. The number of layers is adapted throughout the simulation to maintain homogeneous properties within them, while the water volume in each layer is determined based on the site-specific hypsographic curve.

The thickness of the surface mixed layer is described in terms of a balance of turbulent kinetic energy, comparing the available energy with that required for vertical mixing. The available kinetic energy calculation considers surface wind stress, convective mixing, shear production between layers, and Kelvin–Helmholtz billowing. Mixing in the deeper hypolimnion, is modelled using a constant turbulent diffusivity, or a derivation by Weinstock (1981), in which the diffusivity is calculated as a function of the strength of stratification (described by the Brunt-Väisälä frequency) and the dissipation rate of turbulent kinetic energy.

The general heat budget equation for the uppermost layer considers the balance of shortwave and longwave radiation fluxes and sensible and latent heat fluxes. The effect of heating or cooling by the sediment can additionally be included. The rate of

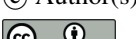



temperature change in each layer is a function of the temperature gradient and the relative area of the layer that is in contact with
the bottom.

Inflows can be characterized by temperature, salinity and other scalar concentration data. Initial mixing is estimated by the inflow entrainment coefficient, which is calculated using a bottom drag coefficient and water column stability characterized by the Richardson number. The inflow Richardson number, in turn, is computed according to the channel geometry and assuming a typically small velocity and Froude number for the drag coefficient (Imberger and Patterson, 1981). Thereafter the inflow is placed
in a layer of neutral buoyancy along the water column. Thus, a new layer is created, with a thickness dependent on the inflow volume.

### 2.2 CE-QUAL-W2

CE-QUAL-W2 (version 3.7) (Cole and Wells, 2006) is a laterally averaged (2DV) model, resulting from the integration in one horizontal direction of the differential equations of conservation of mass, momentum, and energy. It is an open-source Eulerian
model using a structured orthogonal grid that uses a bathymetric map as a geometry input and short-wave radiation, cloud cover, air temperature, dew point temperature, wind speed, wind direction, and precipitation as meteorological forcing. Hydrodynamic output data include water temperature, and longitudinal flow velocity. Laterally averaged models are based on the shallow water equations (Reynolds-Averaged Navier-Stokes equations using the hydrostatic pressure assumption in the vertical, thus neglecting vertical accelerations) and are used for modeling hydrodynamics, water quality, and density stratification in lakes and reservoirs
for which the lateral gradients of those properties are small compared to gradients in the longitudinal and vertical directions. The assumption of lateral homogeneity can be well-suited for describing long and narrow water bodies. The equations are applied in a finite difference grid.

The default turbulence closure model (W2) uses the layer thickness as the mixing length and a formulation for the turbulent viscosity derived by Cole and Buchak (1995). It is also possible to use Nickuradse, parabolic, RNG (renormalization group) and
TKE (turbulent kinetic energy) closure schemes.

### 2.3 Delft3D-FLOW

Delft3D-FLOW (version 4.04.01) (Deltares, 2013), from now on referred to as Delft3D, is a 3D, open source software for simulating the flow and the transport of constituents in water bodies. For the simulation of the hydrodynamics the numerical algorithms behind Delft3D solve the shallow water equations (Reynolds-Averaged Navier-Stokes equations with the consideration
of negligible vertical acceleration – hydrostatic approximation). The simulation of the transport of matter and heat is achieved through the solution of the advection-diffusion equation. The mentioned equations are solved on a structured finite-difference grid using case-appropriate initial and boundary conditions. Delft3D considers user-defined (constant) background viscosities and diffusivities. They represent all forms of mixing that is not resolved through the turbulence closure scheme. In order to calculate the heat exchange between the water surface and the air, five different heat flux model are implemented in Delft3D. Those models
consider the short and long wave radiation balances, evaporation and sensible heat fluxes.

The spatial discretization in the horizontal plane can be performed using curvilinear or rectangular grids, with the former having variable cell size. In the vertical direction a Z- or σ- layer configuration can be employed. In the Z-model, the number of layers is not constant over the basin and varies with local bathymetry.



### 3 Field data

Passaúna Reservoir is a drinking water reservoir located in South Brazil (25.50° S, 49.38° W), which is in operation since 1990. The reservoir is around 11 km long, has 9 km² of surface area and a maximum depth of 16.5 m close to its dam (Sotiri et al., 2019). The main tributary is the Passaúna River with a mean discharge of 2.4 m³ s⁻¹, delivering approximately 75% of the total inflow to the reservoir (Carneiro et al., 2016). Passaúna River enters the reservoir through a small forebay formed at the upstream region of the reservoir due to a bridge (Ferraria Bridge). This forebay has an average depth of 1 m and approximately 0.28 km² of area (Fig.

1). The outflows from the reservoir are the abstraction for the water treatment station, the bottom outlet at the dam (ensuring a minimum discharge of ~0.4 m³ s⁻¹ in the downstream river), and the free overflow spillway. Reservoir bathymetry and its hypsographic curve were obtained from a high-resolution echo-sounder survey (Sotiri et al., 2019). The field measurements described below have been analyzed in detail in (Ishikawa et al., 2021a).

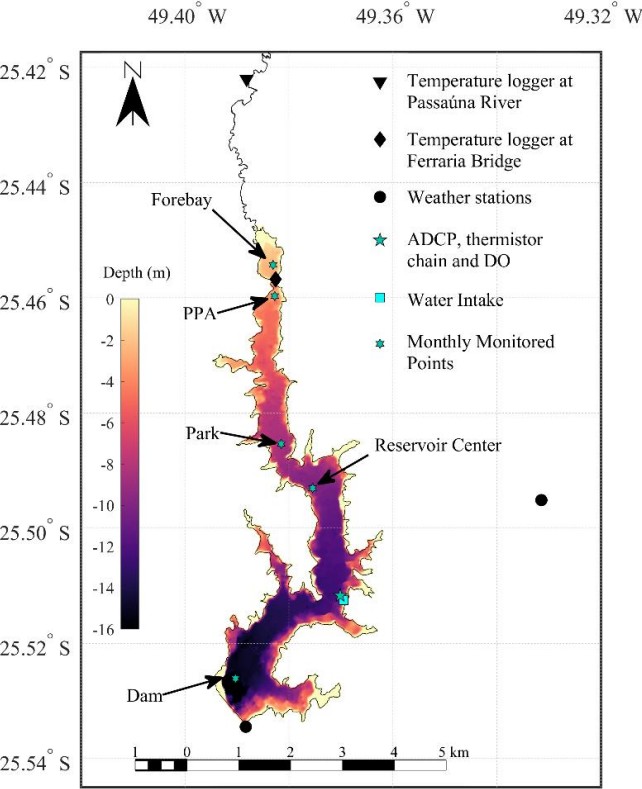

**Figure 1: Bathymetric map of Passaúna reservoir with color representing depth in m in relation to the crest of the spillway (data provided by Sotiri et al. 2019). The inflow of the Passaúna River is in the North, upstream of a bridge forming a forebay. Monitoring stations and main facilities are marked by symbols and explained in the legend.**

### 3.1 Meteorological data

Relative humidity, downwelling short wave radiation, wind (speed and direction at 10 m height), and dew point temperature were

measured at a meteorological station located 4 km east of the reservoir. This station is operated by the Technology Institute of Paraná (TECPAR) and measured every 1 min, here averaged to 1 h. The company operating the reservoir (Sanitation Company of



Paraná, SANEPAR) measured precipitation nearby the dam and, starting from May 2018, they also measured air temperature at the same location (temporal resolution of 10 min, later averaged to 1 h). Starting from this date, the air temperature data were taken from this station. Cloud cover was downloaded from the ERA5 data basis from Copernicus (Hersbach et al., 2018) with hourly

resolution.

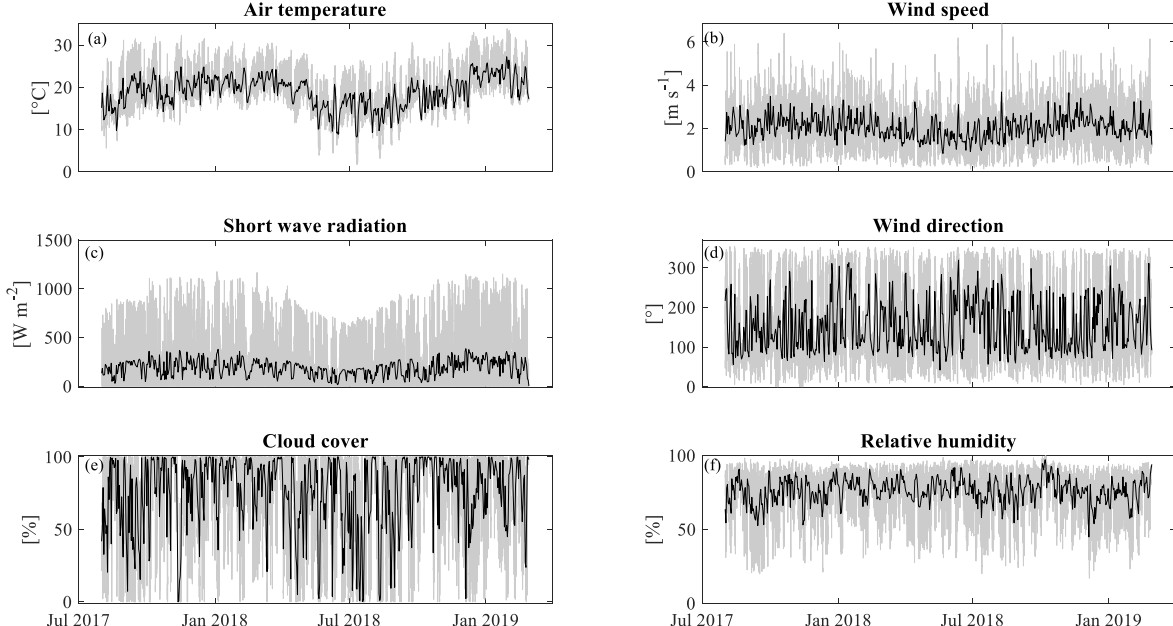

**Figure 2: Time series of meteorological parameters. The gray lines show data with 1 h resolution and black lines are daily averages. Wind direction is measured in degrees clockwise from north.**

Air temperature varied seasonally with a lowest monthly mean value of 14.0 ± 4.3 °C (mean ± standard deviation of hourly time series) in August 2018 and the highest temperature in January 2019 (23.7 ± 3.6°C). Large diel temperature variations followed the daily cycle in shortwave radiation. Wind speed was generally small with a total mean value of 2.0 ± 1.0 m s$^{-1}$. A slight seasonal variation was observed where the lowest monthly averaged wind speed occurred during winter (on June 2018, 1.6 ± 1.0 m s$^{-1}$) and the largest during spring (November 2018, 2.5 ± 1.0 m s$^{-1}$). No seasonal pattern was observed for the remaining parameters (Figure

175    2).

### 3.2 Inflow, outflow and water level

Daily averaged discharge and temperature of the inflows were modelled using the Large Area Runoff Simulation Model (LARSIM-WT, Haag and Luce (2008)). The model was calibrated for the period 2010 to 2013 with a Nash Sutcliffe efficiency of 0.77 (Ishikawa et al., 2021b). In 2018, the model underestimated the peaks of discharges but had good agreement with the baseline.

Simulated water temperature for the year of 2018 had a Nash Sutcliffe efficiency of 0.96.



Starting from March 2018, a temperature logger was installed in the Passaúna River and measured inflow temperature was used instead of the simulated values (sampling resolution of 10 min, here averaged to 1 h). The measurement was made with an accuracy of ± 0.1 °C and resolution of 0.01 °C using a temperature-oxygen sensor (miniDOT, Precision Measurement Engineering, Inc). Water abstraction rate at the intake facility was provided by SANEPAR, measured with an inductive flow meter, and provided at

hourly resolution. The operator also provided reservoir water level measured by an ultrasonic probe in a 30 min temporal resolution. Outflow discharge at the ground outlet and the spillway were calculated based on standard hydraulic structures design equations, according to the structures geometry. The discharge coefficients were adjusted using a few downstream discharge measurements (MuDak-WRM project, Fuchs et al. (2019)), but also considering the overall water balance, where simulated inflows minus calculated outflows should correspond to the measured water level changes.

**3.3 Temperature**

Close to the intake facility, where the water depth is about 12 m, a thermistor chain was deployed from 1 March 2018 to 6 February 2019. The chain was fixed at the bottom and had 11 temperature loggers with 1 m vertical spacing, starting from 1 m above the bed. The loggers (Minilog-II-T, Vemco) measured at a sampling interval of 1 min with a precision of ± 0.1°C and 0.01°C resolution. An additional logger of the same type was placed under the Ferraria Bridge, with the same configuration from 2 March 2018 to 12

August 2018. In addition, temperature profiles were collected with a CTD (Conductivity-Temperature-Depth profiler, Sontek Cast-Away) at five locations along the reservoir (Fig. 1) in February, April, May, June, August, November and December 2018, and February 2019.

**3.4 Flow velocities**

An upward looking acoustic Doppler current profiler (ADCP Signature 1000, Nortek AS) was deployed close to the thermistor

chain (<50 m distance) at the bottom of the reservoir to measure vertical profiles of flow velocity. The device was deployed and recovered for data download and battery replacement several times from 23 February 2018 to 05 February 2019. Its configuration was modified between individual deployments (Table SI 1) to improve the data quality, and also to adjust power consumption (measurement duration) to the monitoring program. Mean values of the three-dimensional flow velocities were recorded along a vertical profile starting at 0.7 m above the bed up to 1.5 m below the water surface with vertical and temporal resolution of 0.5 m

and 5 or 10 min, respectively. High-resolution profiles of vertical velocities were used for turbulence analysis. These profiles covered a depth range of 7.4 m, starting 0.6 m above the bed with a spatial (vertical) resolution of 4 cm and a sampling frequency of 1 or 4 Hz. High-resolution data are not available for the first ADCP deployment.

**4 Model setup**

The simulation period started on 1 August 2017 and ended on 28 February 2019. The first six months were considered as a spin-

up period for the models, it was decided to start the simulations on August when the reservoir was vertically mixed. Therefore, all models started with uniform temperature of 17 °C and water level at 887.01 m.a.s.l. In addition, conservative tracers were implemented to observe the transport of substances from Passaúna River, hence the river had a constant concentration of 1 kg m$^{-3}$ starting from 1 Aug 2018. Specific definitions for each model are presented in Table 1. Each model underwent a manual calibration





procedure, in which the listed coefficients (see Table 1) were modified in order to reduce the mean absolute error (MAE) of the
temperature.

**Table 1: Specification of model parameters and settings. Coefficients are indicated as: calibrated – (default).**

|  | GLM | CE-QUAL-W2 | Delft3D |
|---|---|---|---|
| Horizontal grid | Not applicable | 2 branches<br>Main branch: 72 segments<br>Sec. branch: 5 segments<br>Branches width: ~ 150 m | Curvilinear<br>Cell grids of ~ 40 x 40 m |
| Vertical grid | Mixed layers scheme<br>Maximum number of layers: 500<br>Min layer volume: 0.025<br>* Min – max: 0.1 – 0.5 m | Z – 20 layers<br>0.85 m | Z – 20 layers<br>0.83 m |
| Time step | 3600 sec | 1 sec | 0.1 min |
| Computational time<br>Specifications: Intel®Core™i5-8400 CPU @ 2.80GHz 2.81 GHz | 5 s | 3.69 min | 85.2 h |
| Surface heat flux approach | Longwave radiation: calculated by the model internally from the cloud cover and air temperature. Solar radiation flux: albedo calculation option 4 - sub-daily approximation | **Term by term model**<br>Longwave radiation: calculated by the model internally from the cloud cover and air temperature. Solar radiation flux: albedo is calculated according to solar altitude | **Ocean heat flux model**<br>Longwave radiation: calculated by the model as total net longwave radiation, as a function of cloud cover, relative humidity and air and water surface temperature<br>Solar radiation flux: constant albedo (0.06) |
| Evaporative heat flux approach | Estimated by the vapour pressure differences and wind-driven convection as function of air density and pressure | Estimated by the vapour pressure differences and wind-driven convection with constant empirical coefficients | Estimated from relative humidity, summing the forced (wind dependent) and free convection of latent heat |
| Coefficient for latent heat transfer (-) | 0.002 – (0.0013) | - | 0.0013 – (0.0013) |
| Coefficient for sensible heat transfer (-) | 0.0015 – (0.0013) | 0.47 mm Hg °C$^{-1}$ (Bowen's coeff.) | 0.0013 – (0.0013) |
| Wind coefficient | Bulk aerodynamic transfer coefficient for momentum (-):<br>0.0013 – (0.0013) | Wind roughness Height (m):<br>0.001 – (0.001) | Wind drag coeff. (-):<br>Wind intensity    coef<br>0 – 1.25 m s$^{-1}$    0.003<br>1.25 – 3.00 m s$^{-1}$  0.0025<br>> 3.00 m s$^{-1}$    0.0018 |
| Light extinction coefficient (m$^{-1}$) | 0.85 – (0.50) | 0.50 – (0.45) | 0.85 – (0.85) |
| Heat exchange with the sediment | Neglected | Neglected | Not applicable |
| Turbulence closure model | Option 2: Derivation by Weinstock (1981) whereby diffusivity | $k – \varepsilon$ model | $k – \varepsilon$ model |



| | | | |
|---|---|---|---|
| | increases with dissipation and decreases with increasing stratification | | |
| Horizontal eddy viscosity (m² s⁻¹) | Not applicable | 1.0 – (1.0) | Computed, definition of a background 0.0 – (10) |
| Horizontal eddy diffusivity (m² s⁻¹) | Not applicable | 1.0 – (1.0) | Computed, definition of a background 0.0 – (10) |
| Vertical eddy viscosity (m² s⁻¹) | Kinematic viscosity of water: $1.14 \times 10^{-6}$ | Computed with a definition of a maximum 1.0 – (1.0) | Computed, definition of a background $0.0 - (10^{-6})$ |
| Vertical eddy diffusivity (m² s⁻¹) | Diffusivity of scalars in water due to turbulent mixing: computed | Computed | Computed, definition of a background $0.0 - (10^{-6})$ |
| Bottom friction | Computed | Manning 0.035 | Manning 0.035 |

### 4.1 Boundary conditions

The same boundary conditions were used for all three models, with temporal resolutions according to the availability of data and model requirements. The boundary conditions are: air temperature, relative humidity, downwelling short wave radiation, wind speed, wind direction, precipitation, cloud cover, water level, outflow discharge (for water intake and continuous discharge of ground outlet), and inflow discharge and temperature.

### 4.1.1 Bathymetry and grids

Bathymetric information were interpolated on the grids of Delft3D and Ce-Qual-W2, whereas GLM only used the hypsographic curve. The CE-QUAL-W2 grid was built using a QGIS 3.2 plugin developed by (Bornstein, 2019). It contains a maximum of 20 layers (vertical direction), 2 branches (longitudinal direction) and 82 segments divided among the two branches (Figure 3b).
The Delft3D grid was built using the grid generator of Delft3D (RGFGRID). The resolution is higher in the Forebay-Bridge area in order to better represent the formation of density currents in this region (Figure 3e and f). The refinement of the grid in the
forebay region was made using the RGFGRID module and where needed the bathymetry and grid was edited using the Quickin module for a better representation of the reservoir.

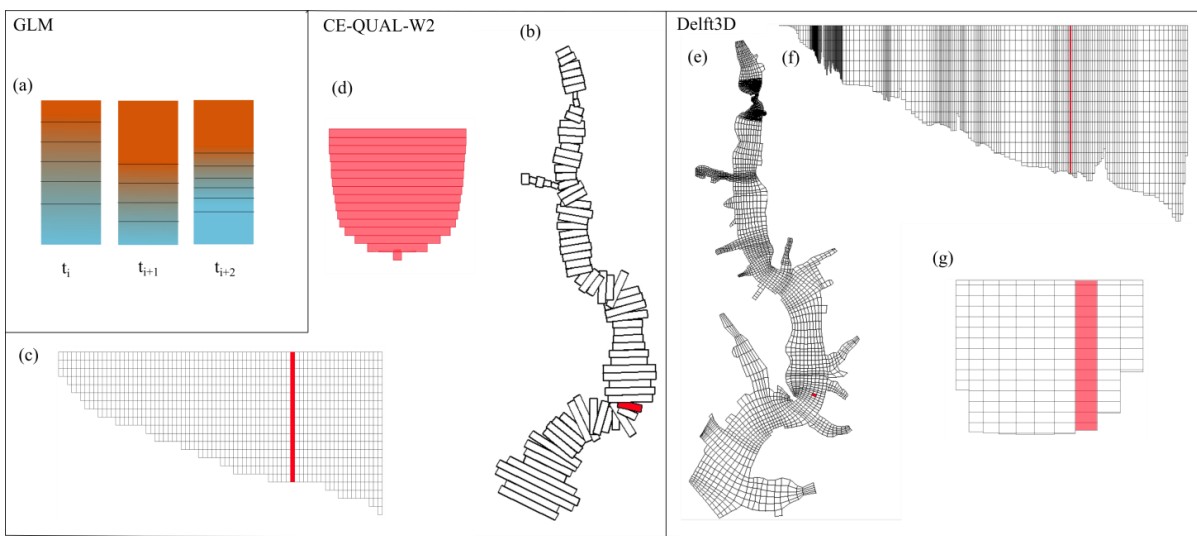

**Figure 3: Overview of grids. (a) Representation of GLM cells (vertical layers) that expand or contract according to mixing, thus changing their total number during simulations. CE-QUAL-W2 grid in (b) top view, (c) longitudinal view, and (d) transversal view. Delft3D grid in (e) top view, (f) longitudinal view, and (g) transversal view. Red background represent the cells that were used for comparison with monitoring data near the intake station.**

### 4.1.2 Inflow

GLM and CE-QUAL-W2 used inflow discharge and temperature with daily resolution, which was required for GLM, while the 3D model used temperatures with 10 min temporal resolution for the Passaúna River after the installation of a temperature logger. In GLM the inflows were divided in 3, the main two tributaries (Passaúna and Ferraria River) and the other 60 minor tributaries as a single discharge, their discharged were summed up and the temperatures averaged. In CE-QUAL-W2 and Delft3D, each tributary was implemented as a single discharge at the closest respective segment / cell. The intrusion depth for CE-QUAL-W2 was defined as the layer having the same density, and for Delft3D it was uniformly distributed over depth.

### 4.1.3 Intake

The intake facility had withdrawal flow rates implemented in daily resolution for the 1D and 2D models, and in hourly resolution for the 3D. The abstraction of water occurred close to the surface, therefore for GLM and CE-QUAL-W2 the abstraction level was set up as 885 m.a.s.l., and the surface cell was defined for Delft3D. The abstraction location was defined at the 2D and 3D model and for the 1D only the level was required.

### 4.1.4 Ground outlet

The ground outlet flow rate was almost constant ($0.44 \pm 0.07$ m$^3$ s$^{-1}$) over the simulation period, as there were only a few gate operations, and small water level variations. This flow rate was abstracted from all models in a daily temporal resolution. Similar to the intake withdrawal, GLM required as additional information the level of the outlet (872 m.a.s.l.). For CE-QUAL-W2 and Delft3D the second deepest cell was selected.



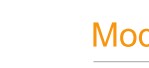 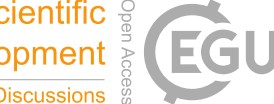

### 4.1.5 Spillway

The spillway was set up as an open boundary in the three models, for GLM and CE-QUAL-W2 its discharge was computed through the following equations:

$$Q_{spillway} = \alpha \Delta h^{1.5} \tag{1}$$

where $Q_{spillway}$ (m³ s⁻¹) is the volumetric flow over the spillway, $\Delta h$ (m) is the difference between water level and spillway crest. For GLM $\alpha$ was calculated as:

$$\alpha = 2/3 C_{Dspill} \sqrt{2g} W_{spill} \tag{2}$$

which depends on spillway width, $W_{spill}$ = 60 m, and associated drag coefficient $C_{Dspill}$ = 0.62, and $g$ is the acceleration due to gravity (m s⁻²), while in CE-QUAL-W2 it was defined as an empirical coefficient = 110.3.

For Delft3D this open boundary was defined as water level dependent, where the measured water levels were applied in temporal resolution of 30 min. The outflow discharge thus depends on the water level and the bathymetry at the open boundary grid cells.

### 4.1.6 Shorelines, bed and water surface

Shorelines and bed are considered as closed boundaries with no-flux condition. For Delft3D and CE-QUAL-W2 a uniform roughness coefficient was specified at the bed (Table 1). Surface heat fluxes are described in Table 1, and wind direction was not used in the 1D model. Precipitation was uniformly distributed over the water surface.

### 4.1.7 Meteorological data

GLM and CE-QUAL-W2 used meteorological data with a temporal resolution of 1 h. For GLM this resolution is equivalent to the minimum time step. For Delft3D, meteorological data with 10 min temporal resolution were used.

## 5 Indices for comparison

To compare model simulations to observations, the following parameters were calculated for the cells being closest to the sampling location. For CE-QUAL-W2 the segment 55 was selected (indicated in Figure 3a, b and c), for Delft3D cell [193, 28] was selected
(Figure 3d, e and f), and for GLM the first 12 m of depth were selected. Simulated values were linearly interpolated to match with the sampling depths, and indices were calculated only the period of interest: 01 Mar 2018 to 28 Feb 2019.

### 5.1 Statistics

The Taylor diagram (Taylor, 2001) provides a concise overview of the performance of models through comparison with observations in terms of standard deviation ($\sigma$), correlation coefficient ($R$), and centered root mean squared error (cRMSE) in one
plot. These parameters were calculated as follows:

$$\sigma = \sqrt{\frac{1}{n-1} \sum_{i=1}^{n} \left(a_i - \bar{a}\right)^2} \tag{3}$$

where $n$ is the number of observations, $a_i$ is one observation and $\bar{a}$ is the mean of all samples (measured or simulated).





$$R = \frac{\frac{1}{n}\sum_{n=1}^{n}\left(s_i - \bar{s}\right)\left(m_i - \bar{m}\right)}{\sigma_s \sigma_m}$$

(4)

where $s$ denotes the simulated value and $m$ the measured.

$$cRMSE = \sqrt{\frac{1}{n}\sum_{n=1}^{n}\left[\left(s_i - \bar{s}\right) - \left(m_i - \bar{m}\right)\right]^2}$$

(5)

Some results are compared in terms of the mean absolute error (MAE):

$$MAE = \frac{1}{n}\sum_{n=1}^{n}\left(s_i - m_i\right)$$

(6)

### 5.2 Mixing and stratification

Following (Ishikawa et al., 2021a), the water column was classified as mixed or stratified based on a threshold of the Schmidt

stability ($S_T$). Days with daily averaged Schmidt stability equal or lower than 10% of the annual maximum Schmidt stability (calculated with the measured data as 16.3 J m$^{-2}$) were considered as mixed, while days with higher $S_T$ were classified as stratified. Schmidt stability was calculated using the software Lake Analyzer (Read et al., 2011), as:

$$S_T = \frac{g}{A_S}\int_{0}^{z_D}\left(z - z_v\right)\rho_z A_z dz$$

(7)

where $g$ (m s$^{-2}$) is the gravitational acceleration, $A_S$ is the reservoir surface area, $A_z$ is the area at depth $z$, $\rho_z$ is the water density

at depth $z$, $z_D$ is the maximum depth, $z_v$ is depth of the reservoir center of volume calculated as $z_v = \int_{0}^{z_D} z A_z dz \bigg/ \int_{0}^{z_D} A_z dz$.

The thickness of the upper mixed layer (UML) was also computed by Lake Analyzer using a threshold for the vertical density gradient, which depends on the density gradient of the entire water column (see Read et al. (2011) for details).

### 5.3 Temperature

Due to the dynamic mixed layers of GLM, its results were linearly interpolated for each 0.5 m over depth. To calculate errors the measured temperatures correspondent to the timestamp of simulation results were selected, and simulation results were linearly interpolated to match with the observations over depth. This procedure was followed for the three models.

### 5.4 Flow velocities

CE-QUAL-W2 provided one horizontal velocity components (longitudinal velocity), being positive in the downstream direction

and negative in the upstream direction. To compare flow velocities, the longitudinal component of Delft3D and the measurements were computed by aligning the flow in the same direction as the 2D model, thus the transversal velocity component was not considered in our analysis.





## 6 Results

### 6.1 Water level / water storage

At the end of the simulation period, all three models presented a lower water level than the measured. The last measurement was 886.81 m.a.s.l, the closest value simulated was in Delft3D with 886.49 m.a.s.l., which was also the one with the lowest error (MAE = 7.4 cm). GLM simulated a final water level of 886.44 m.a.s.l. and CE-QUAL-W2 estimated 886.36 m.a.s.l, with respective MAE of 10.5 cm and 10.8 cm, respectively (Fig. SI 1a).

    The largest discrepancies in water level occurred when it raised over the spillway crest. GLM and Delft3D had water above the
crest for a longer period than observed, and their levels kept being larger than the measurements until a sharp increase in October 2018, which none of the models reproduced. Total spillway discharge had its largest volume in CE-QUAL-W2: $2.93 \times 10^7$ m$^3$, GLM had a spillway volume as the 2D model $2.87 \times 10^7$ m$^3$, and Delft3D simulated 3.7 % less spillway discharge than CE-QUAL-W2 ($2.83 \times 10^7$ m$^3$) (Fig. SI 1b).

    Evaporation in all models were in the same order of magnitude, but significantly different (one-way ANOVA test with $p$-value = 
$5 \times 10^{-17}$). The 1D, 2D and 3D models estimated daily mean evaporation rates of $2.9 \pm 1.3$ mm day$^{-1}$, $2.7 \pm 1.0$ mm day$^{-1}$ and $3.4 \pm 1.4$ mm day$^{-1}$, respectively. Comparing the volumes due to evaporation with the reservoir volume ($7.0 \times 10^7$ m$^3$), over the year GLM lost due to evaporation the equivalent to 12.0 % of the reservoir volume, CE-QUAL-W2 with the lowest evaporation rate lost 11.0 % and Delft3D 14.3 %.

### 6.2 Temperature

**6.2.1 Vertical profile at the intake region**

    From the measurements made with the thermistor chain, it was observed that the reservoir was thermally stratified at the beginning of the monitoring period (end of summer). The first autumn overturn took place in mid-April, but after a few days the reservoir became stratified again. This dynamics of mixing and stratification repeated several times throughout autumn and winter, characterizing a warm polymictic mixing regime (Lewis Jr, 1983; Ishikawa et al., 2021a). Persistent thermal stratification
developed in spring, and retained over summer.

    The observed seasonal pattern of stratification and mixing was reproduced by all three models (Figure 4). At the water surface, simulated temperatures were highly correlated with observations with comparable correlation coefficients (>0.99) for all three models (Figure 5b). The net surface heat fluxes simulated by the models were not statistically different ($p$-value = 0.14). Observed water surface temperature was $21.6 \pm 3.5$ °C (mean value and standard deviation) for the whole period. In the simulations, surface
temperature was $22.2 \pm 4.0$ °C in GLM, $21.1 \pm 3.7$ °C in CE-QUAL-W2, and $22.4 \pm 3.7$ °C in Delft3D. With increasing depth, the error increased and the correlation between measured and simulated temperatures decreased (Figure 5). In the deepest layer, temperature was on average $19.1 \pm 2.0$ °C, while GLM, CE-QUAL-W2 and Delft3D simulated $18.6 \pm 2.2$ °C, $18.2 \pm 2.3$ °C and $19.3 \pm 2.3$ °C, respectively.



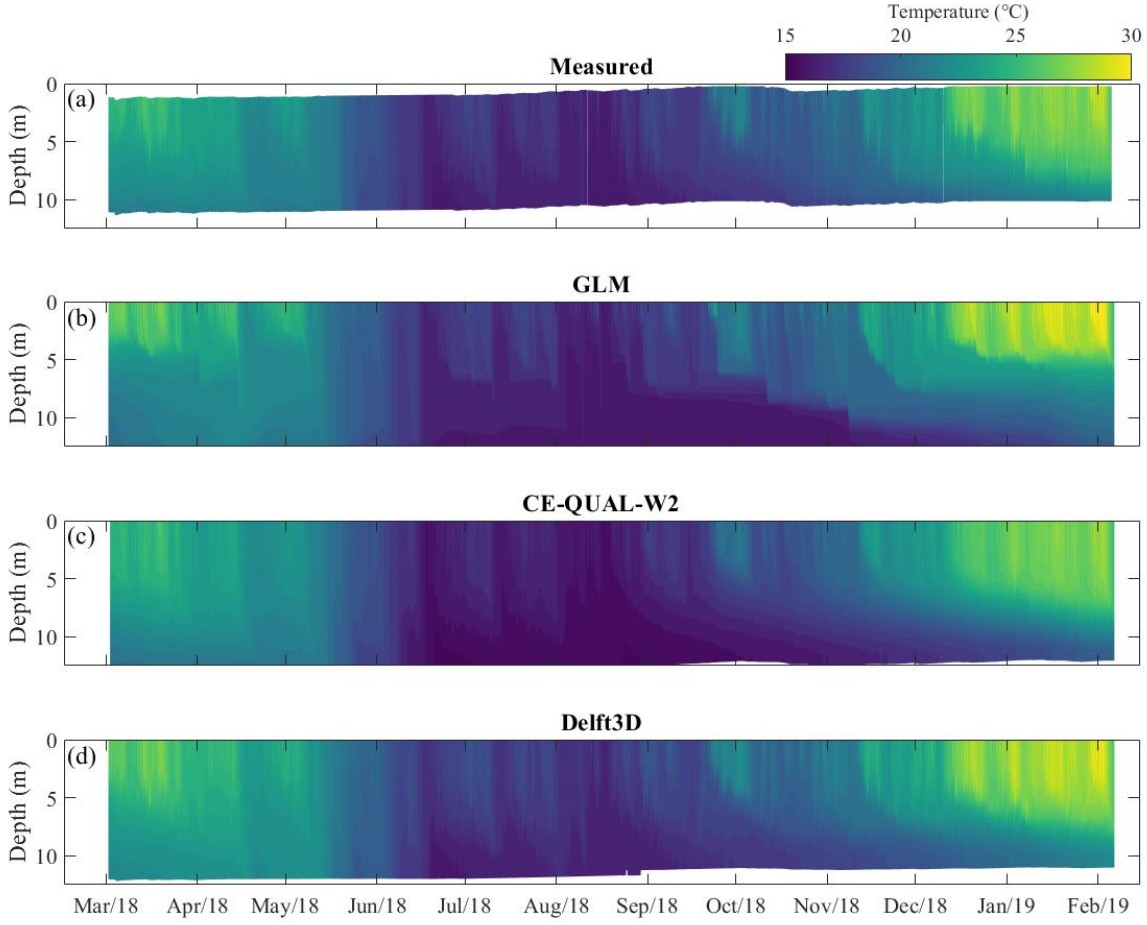

**Figure 4: Contour plots of vertical temperature profiles at the location of the thermistor chain near at the Intake region of the reservoir (Fig. 1). (a) Measured temperature in Δt = 1 h and Δz = 1 m, (b) simulation result of GLM (Δt = 1 h and Δz = 0.5 m), (c) simulation result of CE-QUAL-W2 (Δt = 1 h and Δz = 0.85 m), (d) simulation result of Delft3D (Δt = 1 h and Δz = 0.83 m).**

In the 2D model, errors increased rather continuous with increasing depth, showing maximum cRMSE of 0.6 °C at around 10.5 m

depth. Meanwhile, GLM and Delft3D showed largest errors around the middle of the lower half of the water column, GLM at a

depth of 6.7 m with cRMSE of 0.94 °C, and Delft3D at around 8.6 m depth with an error of 0.6 °C.

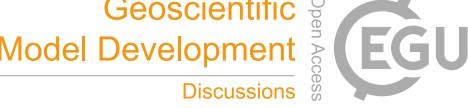

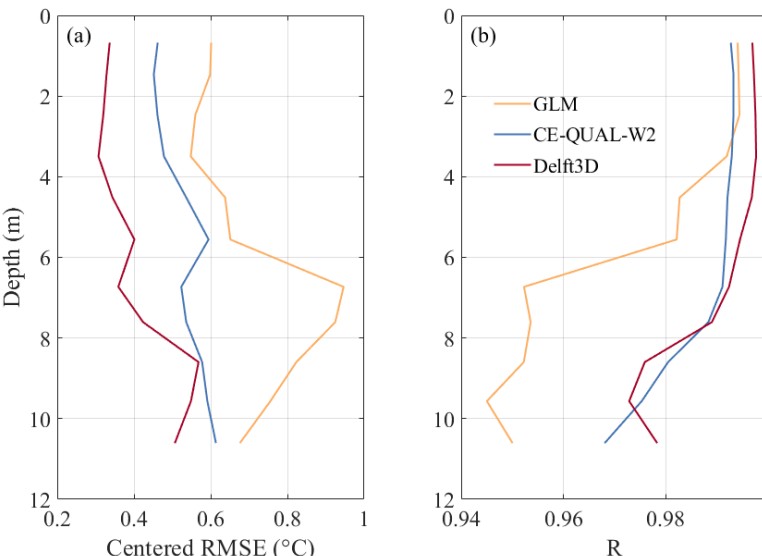

**Figure 5: Performance of temperature simulations from GLM, CE-QUAL-W2 and Delft3D for the intake region.** (a) **shows the centered root mean squared error (cRMSE) and** (b) **correlation coefficient (R) along depth for each model.**

The average water temperature in the GLM simulations was about 0.5°C warmer at the surface and colder at the bottom, which lead to stronger thermal stratification than in the observations. CE-QUAL-W2 simulated lower temperatures at the surface and bottom (− 0.5 °C and − 0.9 °C, respectively), and Delft3D estimated warmer temperatures at surface and bottom (+ 0.8 °C and + 0.2 °C, respectively).

According to the classification based on Schmidt number, Passaúna Reservoir was mixed on 95 d out of 343 d of the monitoring
period, with the longest continuous period of mixing from 08-May to 20-Jun 2018 (Figure 6a). In GLM, stratification was generally more stable and the reservoir was classified as mixed only on 68 d. Periods with homogeneous temperature were shorter and discontinuous, and the last mixing event in early September was not resolved by the model. The simulated Schmidt stability was strongly correlated with those estimated from observations ($R^2$ = 0.96, $p$-value = $4 \times 10^{-290}$), however it was overestimated by a factor of 1.64 on average (Figure 6b). CE-QUAL-W2 provided the closest match of the number of mixed days with observations
(95 d). Due to the lower simulated bottom temperature, the Schmidt stability was overestimated by a factor of 1.19 on average, but simulations and observations were highly correlated ($R^2$ = 0.90, $p$-value = $1 \times 10^{-228}$, Figure 6c). During the mixed season, the intermittent stratification was attenuated in the 2D model, and the mixed periods were slightly longer. Delft3D had the best correlation and a lower overestimation of Schmidt stability than GLM ($R^2$ = 0.97, $p$-value = $4 \times 10^{-311}$, factor of 1.12), however the number of mixed days in the simulations was underestimated by about 25 % (72 mixed days) (Figure 6c).





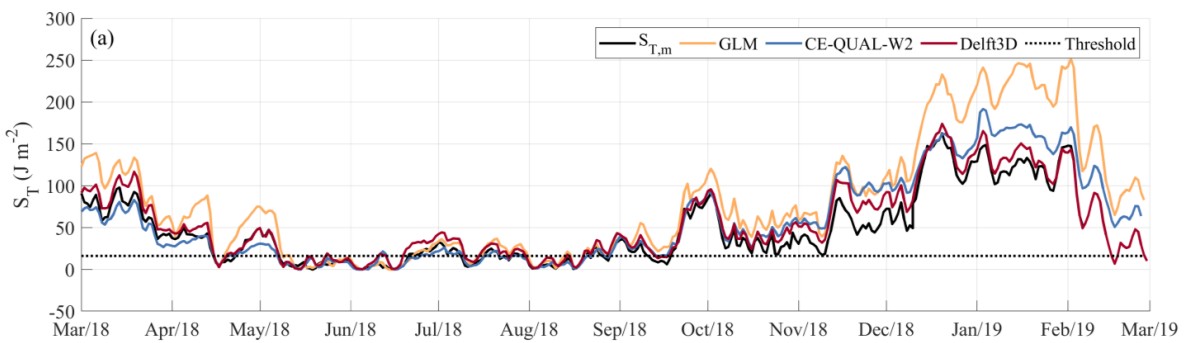

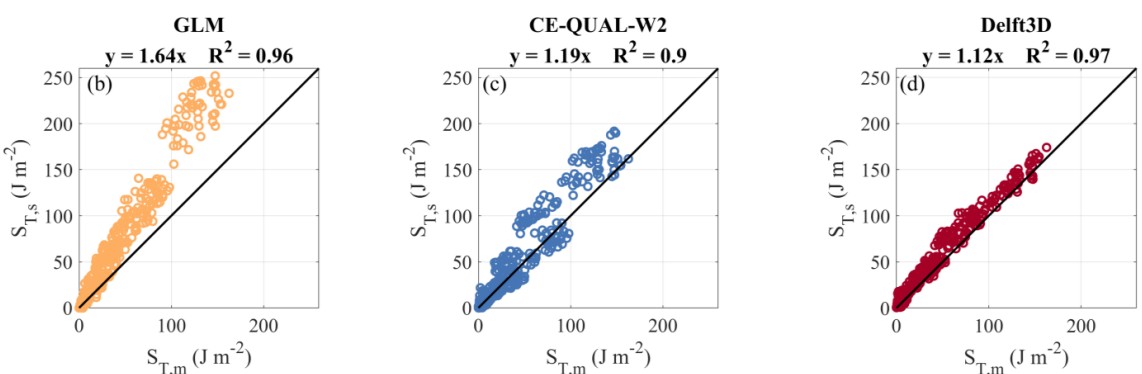


**Figure 6: (a) Time series of daily-averaged Schmidt stability ($S_T$) estimated from observed ($S_{T,m}$) and simulated ($S_{T,s}$) temperature stratification. The dotted line marks the threshold ($S_T = 16.3\ J\ m^{-2}$) used to classify mixed and stratified conditions. Comparison between $S_T$ estimated from measurements and simulation results, linear regression with zero intercept equations and coefficient of determination ($R^2$) are provided as panel title for (b) GLM, (c) CE-QUAL-W2 and (d) Delft3D.**

The upper mixed layer depths (UML) estimated from measurements was compared to UMLs estimated for each model, and all of them presented poor coefficient of determination ($R^2 < 0.6$) for linear regressions (1D: $R^2 = 0.32$ and $p$-value $= 2 \times 10^{-91}$, 2D: $R^2 = 0.38$ and $p$-value $= 1 \times 10^{-100}$, and 3D: $R^2 = 0.54$ and $p$-value $= 1 \times 10^{-129}$ – Fig. SI 3). GLM and Delft3D presented rather thinner UMLs, whilst CE-QUAL-W2 had a larger variance, ranging between deeper and shallower UMLs.

Taylor diagram calculated for temperature simulations throughout the entire period and all depths demonstrate that the three models had good correlations ($> 0.95$) and similar standard deviations of residuals (all models had a standard deviation lower than 0.5 °C for residuals of the difference between measured and simulated temperature). The cRMSE had the most significant differences between models, with Delft3D as the closest to observations (0.50 °C), followed by CE-QUAL-W2 (0.56 °C) and GLM (0.84 °C).



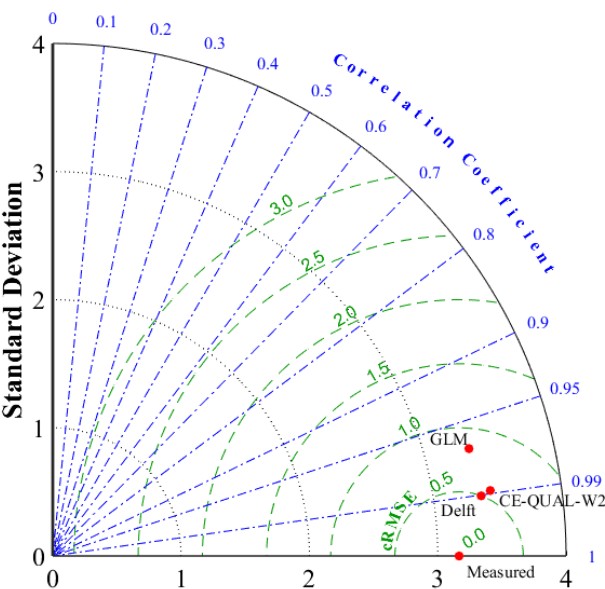

**Figure 7: Taylor diagram of the total simulated period of temperature profiles at the intake region. Green lines indicate centered root mean square error (cRMSE) isolines. Angular coordinate, in blue, represents the correlation coefficient (R). Standard deviation (black dashed line) is represented in radial coordinate with the reference measured data in center. Measured is the baseline where correlation is 1 and cRMSE is zero. The red dots represent model performance.**

### 6.2.2 Longitudinal temperature variations

To compare temperature simulations along longitudinal cross-sections of the reservoir, CTD profile measurements were interpolated and compared with simulations of the 2D and the 3D model (Fig. 8). The models were capable of reproducing the different temperature distributions during the sampling dates. In February 2018 and 2019, the reservoir was stratified in the upstream region with a growing UML along its longitudinal axis. During the remaining surveys in August, November and December 2018, the reservoir showed different patterns of vertical stratification with only minor longitudinal variations. Similar to the mean temperature analyzed above, CE-QUAL-W2 had colder temperatures while Delft3D had higher temperatures, and both had a comparable strength in stratification and were in agreement with the measurements.



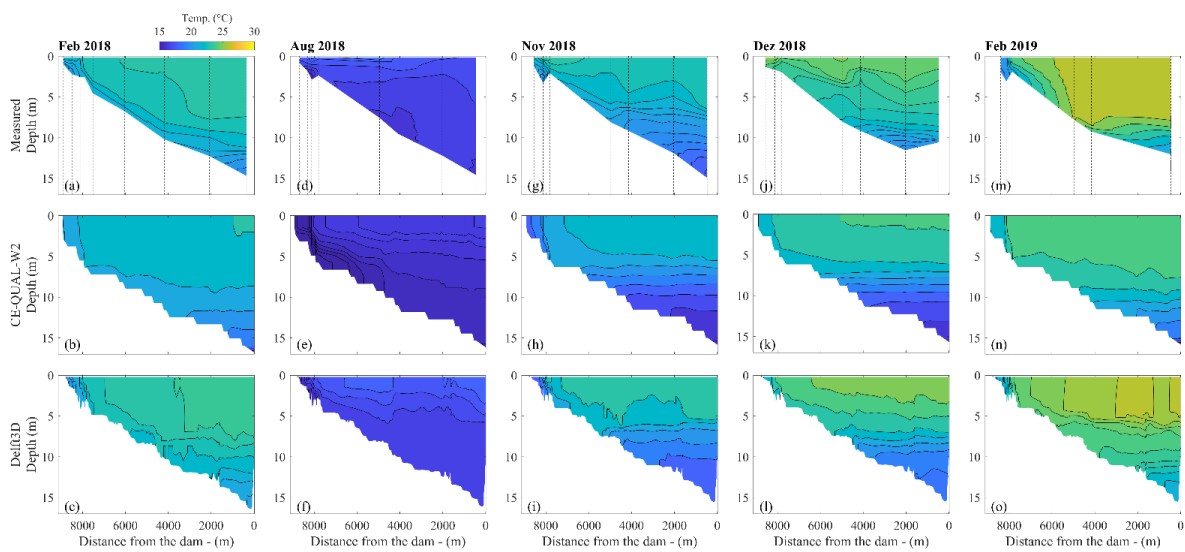

**Figure 8: Contour plots of temperature along a longitudinal cross section of the reservoir from the forebay to the dam. Each column represents one sampling campaign, the first line shows the measurements (interpolated from vertical CTD profiles at locations marked by the dashed vertical lines). The second line are the simulation results of CE-QUAL-W2 for which we show every segment of the grid and all the respective depth cells. The third line of panels shows simulation results from Delft3D, a way along the talweg of the reservoir was drawn by selecting several grid cells, the entire depth of each cell was used.**

### 6.3 Hydrodynamics

#### 6.3.1 Flow velocities

The total averaged horizontal flow velocity (the magnitude of the horizontal velocity components averaged in time and over depth) was around 2 cm s$^{-1}$. Following the analysis of hydrodynamics in Passaúna Reservoir in Ishikawa et al. (2021a), flow velocities larger than 3.5 cm s$^{-1}$ (~ 90$^{th}$ percentile) were defined as currents. They were forced by wind and had the upper part of the water column flowing towards an opposite direction as the lower part (see Figure 10a, b, c and d). The currents, and consequently the total averaged flow velocity, were significantly ($p$-value = $1 \times 10^{-181}$, with null hypotheses of both having a similar distribution) more frequent and more intense during stratified periods when compared to mixed periods. The same analysis, only considering the magnitude of the longitudinal component, was made for the 2D and 3D simulation results (Table 2).

**Table 2: Comparison of magnitude of longitudinal flow velocities (mean ± standard deviation) between measurements and simulations. Measurements and processing are described in Ishikawa et al. (2021a)**

| | Period | Currents (cm s$^{-1}$) i.e. mag. of long. vel. > 90$^{th}$ percentile | Total longitudinal vel. (cm s$^{-1}$) i.e. all mag. of long. vel. | Relative occurrence of currents |
|---|---|---|---|---|
| **Measured** 90$^{th}$ percentile: 3.1 | Mixed | 4.0 ± 0.7 | 1.3 ± 1.0 | 6.4 % |
| | Stratified | 4.4 ± 1.2 | 1.5 ± 1.3 | 10.8 % |
| | Total | 4.3 ± 1.2 | 1.5 ± 1.3 | |
| **CE-QUAL-W2** 90$^{th}$ percentile : 1.8 | Mixed | 2.4 ± 0.6 | 0.9 ± 0.8 | 14.9 % |
| | Stratified | 2.4 ± 0.7 | 0.7 ± 0.7 | 9.0 % |

| | | | | |
|---|---|---|---|---|
| MAE: 1.7 cRMSE: 2.1 | Total | 2.5 ± 0.8 | 0.8 ± 0.8 | |
| **Delft3D** 90th percentile: 2.4 MAE: 1.3 cRMSE: 1.7 | Mixed | 3.3 ± 1.0 | 0.9 ± 0.9 | 7.3 % |
| | Stratified | 3.5 ± 1.1 | 1.1 ± 1.1 | 10.7 % |
| | Total | 3.5 ± 1.1 | 1.1 ± 1.0 | |


For the total period and all depths, CE-QUAL-W2 had a cRMSE = 2.1 cm s$^{-1}$ and a negative correlation coefficient with observations ($-0.04$, $p$-value = $4 \times 10^{-40}$), while Delft3D had cRMSE = 1.7 cm s$^{-1}$ and correlation coefficient of 0.50 ($p$-value = 0). The two models had errors in the same order of magnitude, but the simulations of the 2D model had a lower standard deviation (0.8 cm s$^{-1}$), while the 3D simulations had a standard deviations closer to the observed value (1.4 cm s$^{-1}$, being the observed 1.9

cm s$^{-1}$). Both models showed the largest errors at the surface, where the 3D model is closer to the observations than the 2D model (Figure 9). In contrast to the temperature simulations, the simulated flow velocities had the smallest errors near the bottom. Despite the comparable magnitude of cRMSE of both models, the correlation between simulated and observed velocities differed remarkably (Figure 9b). While it was generally low (<0.1) and fluctuated around zero along the water column for CE-QUAL-W2, it varied between 0.4 and 0.6 for Delft3D.

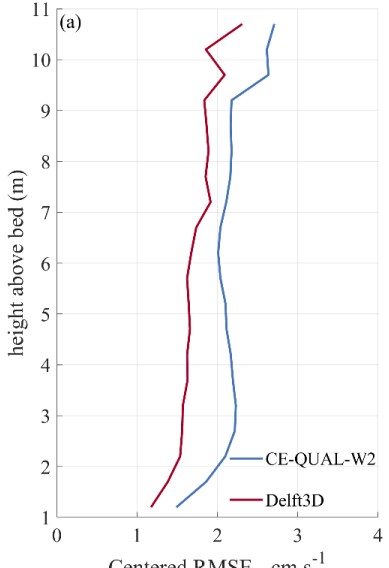
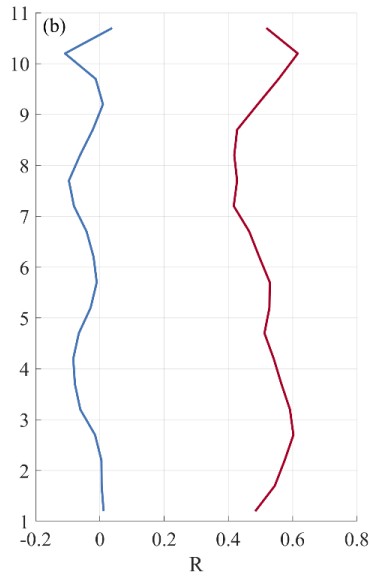


**Figure 9: Performance of longitudinal flow velocities simulations from CE-QUAL-W2 and Delft3D for the intake region.** (a) **shows the centered root mean squared error (cRMSE) and** (b) **correlation coefficient (R) along depth for each model.**

The longitudinal velocities of CE-QUAL-W2 had a 90th percentile of 1.8 cm s$^{-1}$ and a MAE of 1.7 cm s$^{-1}$, its total mean value was

0.8 cm s$^{-1}$ and wind influence on formation of currents is possible to be observed (Figure 10a, b, e and f). The occurrence of currents differed between mixed and stratified conditions and were statistically different ($p$-value = 0.01). Their relative occurrence was larger during mixed conditions, the opposite from observed. In general the simulated flow velocities were lower than observed velocities.


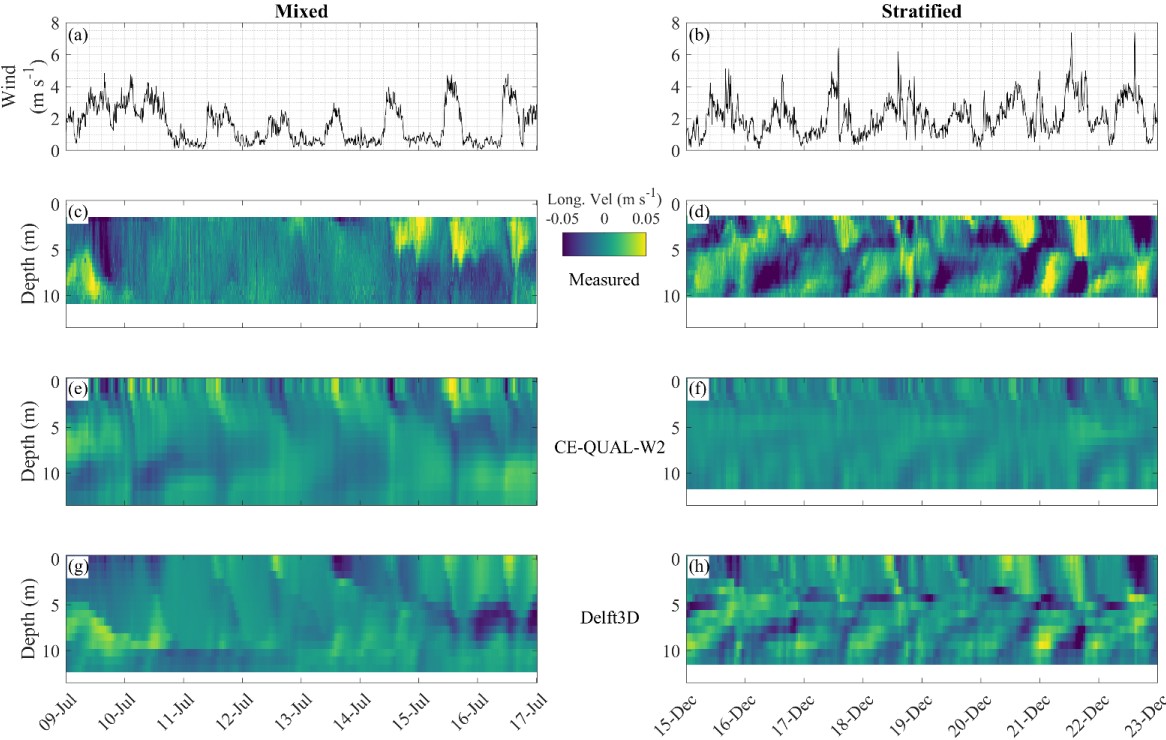

**Figure 10:** (a) and (b)**: Time series of wind speed for two selected periods during mixed (09-Jul-2018 to 17-Jul-2018) and stratified (15-Dec-2018 to 23-Dec-2018) conditions. The other panels show contour plots of the longitudinal flow velocities at the monitoring site (positive values correspond to downstream flow):** (c) and (d) **are observations made by the ADCP,** (e) and (f) **are simulation results from CE-QUAL-W2,** (g) and (h) **are simulation results from Delft3D.**


For Delft3D, MAE was 1.3 cm s$^{-1}$, and its longitudinal velocities were in general lower than observed with a total average of 1.1 cm s$^{-1}$ and 90$^{th}$ percentile of 2.4 cm s$^{-1}$. As in the observations, the occurrence of currents were significantly different ($p$-value = $3.11 \times 10^{-9}$) between mixed and stratified conditions. The simulated currents presented clear opposing directions of flow between in upper and lower depth (Figure 10g and h). In addition, their relative occurrence was within 1% difference from the observed.

### 6.3.2 Turbulence

Only Delft3D provided simulated energy dissipation rates ($\varepsilon$). The estimation of $\varepsilon$ based on measurements is described in Ishikawa et al. (2021a), and due to the limited measurement range of the high resolution mode of the ADCP, estimations were only made up to 8 m height above the bed.

Observed energy dissipation rates were basically the same during mixed and stratified conditions (respective log averages along depth and time: 7.5 and $5.5 \times 10^{-10}$ W kg$^{-1}$). In Delft3D, $\varepsilon$ was approximately one order of magnitude larger at mid depth under



mixed conditions. Log-averaged dissipation rates for the depth range with observations was $8.0 \times 10^{-10}$ W kg$^{-1}$ under mixed, and $1.0 \times 10^{-10}$ W kg$^{-1}$ under stratified conditions (Figure 11a and b)

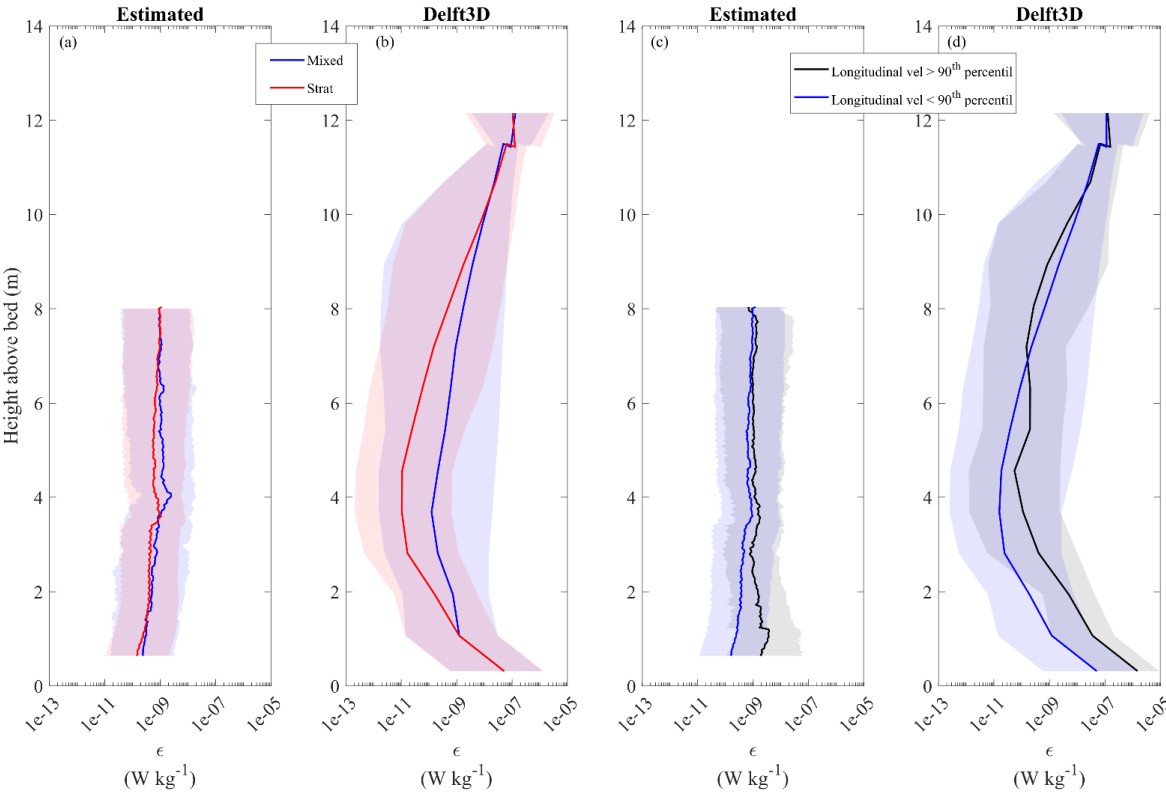

**Figure 11: Solid lines show depth profiles of log-averaged energy dissipation rates ($\varepsilon$). Background shades mark the range between the 5th and 95th percentiles of the temporal variations. (a) and (b) are estimated and simulated (Delft3D) dissipation rates separated between mixed and stratified conditions. (c) and (d) are estimated and simulated dissipation rates divided in periods of longitudinal flow velocities magnitudes exceeding (>) or being smaller ( <) than their 90th percentile.**

While simulations from Delft3D had log averaged profiles with higher $\varepsilon$ towards the bed, the estimations only had the same trend

during the presence of currents (magnitude of longitudinal velocities > 90th percentile). The increase for the estimations started around 3 m above the bed, for simulations the flow velocities under the current threshold also had $\varepsilon$ increasing towards the bottom, and in the presence of currents $\varepsilon$ started to be larger at around 6 m above the bed (Figure 11c and d).

### 6.3.3 Density currents / Substance transport

Substance transport in models was analyzed by continuous addition of a conservative tracer to the inflowing water. Tracer
concentrations at the intake region were assessed to observe substance transport from the Passaúna River to the monitoring site through vertical profiles of the maximum daily tracer concentration. For GLM the entire depth was evaluated (17 m), but it was defined at 12.5 m when maximum tracer concentration was deeper than that, while for CE-QUAL-W2 and Delft3D the closest cell to the station was selected (~ 12 m water depth). The models had similar overall distributions until August 2018 (Figure 12). The



tracer transport changed from interflows to underflows (or to deeper interflows) after the first autumn overturn, with differences
at the depth of daily averaged maximum concentrations.

Underflows and interflows at greater depths were predominant in autumn / winter and overflows and interflows closer to the surface
were more frequent in spring and summer. GLM predicted interflows with the maximum concentrations at shallower depths than
the other models from March to mid-April. After this time, GLM and Delft3D simulations showed underflows more frequently,
while CE-QUAL-W2 results had the interflows moved to deeper regions and presented less underflows when compared to the 1D
and 3D models. Starting from August 2018, the maximum concentration of the tracer showed different patterns in each model. In
the 1D simulations, underflows persisted until the middle of October and after that interflows formed at ~ 5 m depth. In the 2D
simulations, the inflow formed interflows at a slightly larger depth (~ 7 m), while maximum tracer concentrations were widely
scattered over the upper water column in the 3D simulations, with their lower bound following the 2D simulations (Fig. 12).

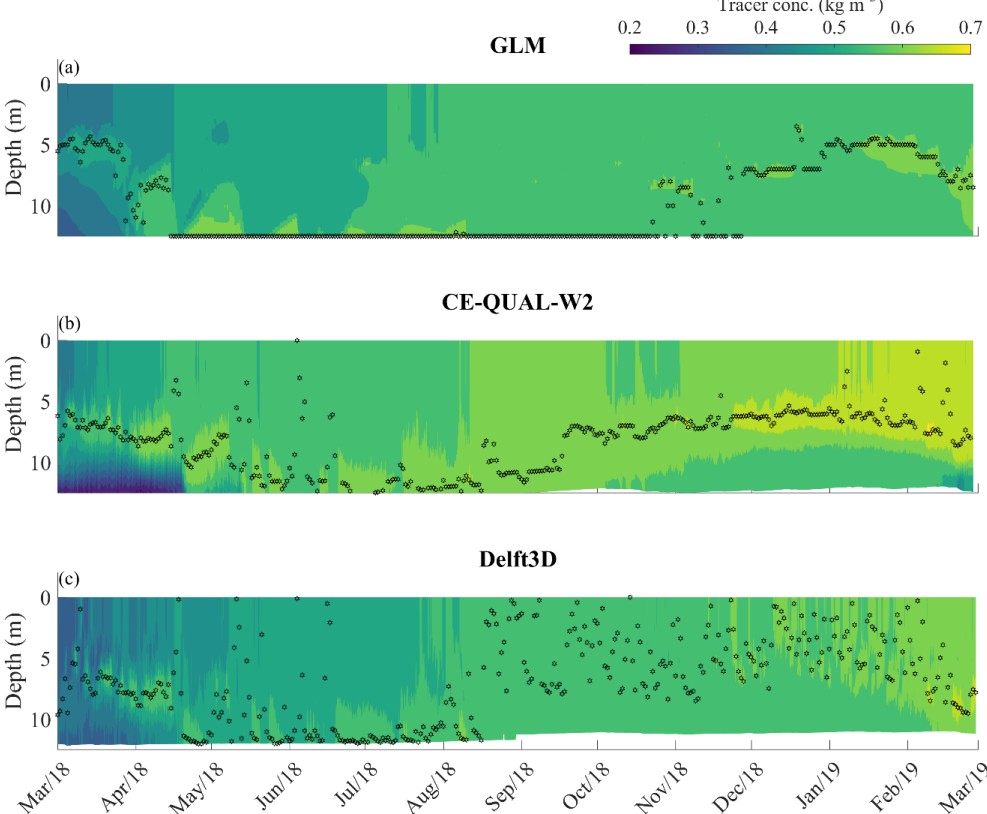

**Figure 12: Contour plots showing simulated time series of tracer concentration along water depth at the intake region obtained from** (a)
**GLM,** (b) **CE-QUAL-W2 and** (c) **Delft3D simulations. Black markers indicate the depth of the maximum daily-mean concentration. The
tracer was introduced continuously in the Passaúna River inflow with a constant concentration of 1 kg m$^{-3}$.**

We calculated the relative frequency of occurrence of each flow path by assigning overflows when the maximum concentration
was at the water surface (uppermost depth cell), underflows when it was at the bottom (lowest depth cell, or deeper than 12 m for
GLM), and interflows otherwise. For GLM there were no overflows, while for most of the time underflows were observed (57.6





%) and interflows for the remaining time. For CE-QUAL-W2 and Delft3D interflows were the most frequent: 75.1 % for the 2D and 67.1 % for the 3D, overflows and underflows were almost equally distributed for Delft3D, respectively 14.9 % and 18.0 %, and CE-QUAL-W2 simulated more frequent underflows (21.4 %) than overflows 3.5 %.

## 7 Discussion

### 7.1 Water storage


Regarding the water balance, all models had errors of comparable magnitude: 1, 2 and 3D had errors in stored water volume −4.4 %, − 4.9 % and − 3.7 %. The error in terms of water level similar for all models (~10 cm), and is in the range of errors reported in the literature (e.g. Dai et al., 2013; Jeznach et al., 2014; Chen et al., 2016; Bueche et al., 2020). GLM had a constant water level from Jan to May 2018, corresponding to the maximum level defined by the hypsographic curve due to the spillway crest elevation.

CE-QUAL-W2 had lower water levels and a larger discharge over the spillway. Both models calculated the discharge by empirical equations. Delft3D was forced by the measured water level as an open boundary at the spillway location. For water level elevations higher than the bathymetry at the outflow cells, water will leave the domain. For water level elevations lower than the bathymetry at the outfall cells no water should flow at all. However, as the bathymetry at the outflowing cells could not reproduce the spillway geometry, periods of water flowing into the domain were observed, which is obviously an artifact. Despite similar results in terms

of storage volume, discharges through the spillway and evaporation differed among models, with Delft3D having the largest evaporation loss and CE-QUAL-W2 the lowest. The differences can be attributed to the differences in how the models describe and implement the boundary conditions. We cannot affirm which model is most precise, because the modelled processes (e.g. evaporation or flow over spillway) were not directly measured. In addition, measured data are also associated with uncertainties. There is an underestimation of peaks of inflow discharge from LARSIM-WT, and poor data accuracy on outflows of the bottom

outlet at the dam and the spillway discharge, which were important parameters that contributed for the discrepancies of the water balance of the reservoir. Thus, care has to be taken when defining boundary conditions in all models, and the first step should always be to check water balance and flows at the boundaries.

### 7.2 Temperature

All three models simulated the dynamics of temperature stratification in reasonable agreement with observations. Resulting errors

are in the same order of magnitude as values reported in other model applications (e.g. Bruce et al., 2018; Weber et al., 2017; Kobler et al., 2018; Mi et al., 2020; Chanudet et al., 2012; Dissanayake et al., 2019).

Stratification is mostly driven by heat exchange associated with absorption of solar radiation, net longwave radiation, evaporation, precipitation, and sensible heat transfer at the water-air interface. Despite the different approaches for the heat flux and the use of different coefficients (see Table 1), daily averages of net surface heat flux of the three models had no significant difference among

each other. Other processes that influence the temperature stratification are inflows, surface runoff, groundwater inflow, and heat exchange with the sediment (Wetzel, 2001). Out of those, only river inflow temperatures were known, and implemented in all models. The others were considered as negligible. Regarding the heat exchange with sediment, Stepanenko et al. (2013) showed that it did not have significant influence on simulations of bottom water temperature of a shallow lake for a comparable temperature range as observed in Passaúna Reservoir. Even though the errors were of comparable magnitude, the model simulations of thermal





stability differed among the models, with underestimated temperature in the 2D simulations and overestimated temperature in the 3D simulations. We assume that the differences were related to model dimensionality and the parameterization of vertical mixing and inflows.

The 1D model imposes the largest simplification by neglecting horizontal variations in flow and water temperature, even though an initial mixing of inflows due to entrainment is parameterized in the model. The lower temperatures in the deeper layers can be

explained by the cold inflow temperatures combined with the inability of reproducing the enhanced heat exchange of the inflowing water with the atmosphere in the shallow forebay. In addition, the surface temperature was overestimated by GLM, which explain the larger number of stratified days and increased thermal stability. Another factor that can potentially contribute to errors is the selection of the first 12 m in depth for comparison with measurements, thus excluding the bottom boundary layer of the 1D vertical grid. On the other hand, the 2D and 3D models had a better representation of the strength if vertical density stratification, despite

having overall divergent results – respectively colder and warmer temperatures than measured. These differences can be at least partially explained by the calibration process, which becomes more difficult for increased dimensionality. Due to the short computational time required for 1D models, it is possible to perform repeated runs with varying calibration parameters (e.g. light extinction coefficient) and to improve agreement with data during model calibration. Moreover, tools for automated model calibration are available (Bueche et al., 2020).

CE-QUAL-W2 estimated 92 days with mixed conditions, which is in closest agreement with the observations (95 d), but their temporal dynamics differed from observations especially during winter. The intermittency between mixing and stratification was more frequent, mixed periods were longer from June to August and the last mixing event shorter. In the Delft3D simulations, the shorter mixed periods are missing, which reduced the total duration of mixed conditions to 78% of the observed.

### 7.3 Longitudinal processes

The simulated temperature distribution along the longitudinal cross section of the reservoir, only possible to evaluate with 2D and 3D models, had good agreement with observations. Hence, we can assume that in general the processes at the transversal direction are of minor importance for the stratification at the Passaúna Reservoir. This is further supported by the good agreement between both models for the simulated tracer transport from the river inflow to the water intake station. Nevertheless, the tracer analysis showed how the differences in temperature simulated by each model also affected the inflow pathways.

These results highlight the advantages of CE-QUAL-W2 and Delft3D, as they are capable to represent the observed longitudinal gradient, especially considering the inflow region, where colder river water flows downwards as an underflow. This underflow is responsible for transporting not only cold water, but dissolved nutrients and suspended sediment over long distances into the deeper parts of the reservoir. Those dynamics are represented in GLM only in a very simplified manner, explaining the weaker performance of GLM with respect to water temperatures at higher depths (Figure 5).

Tracer dynamics observed with GLM complies with the hypothesis that the lower temperature at the bottom was caused by inflow pathways of mostly underflows, because of the absence of the forebay. Fenocchi et al. (2017) demonstrated that, in order to reproduce the thermal response to inflows in a subalpine lake with GLM, it was necessary to use an impractical coefficient of light extinction. The general colder temperatures simulated by CE-QUAL-W2 placed the inflow at larger depths and confined them in layers. This behavior was observed because the longitudinal flows were located below the UML, thus the higher concentrations of

tracer especially after September (Figure 12). For Delft3D the opposite was observed, the density currents were within the UML, which diluted the tracer concentration and the depth of its maximum was strongly variable over the last 6 months. The travel time





of the tracer was evaluated by identifying the first time that the tracer concentration was larger than $10^{-3}$ kg m$^{-3}$ at the intake after the release of the tracer at Passaúna River. The transport in CE-QUAL-W2 was faster with a time travel of 2.2 days and 3.5 days in Delft3D, which can be associated with the higher tracer concentrations of the 2D model. This information is important for

management of reservoirs during spilling accidents (e.g. Jeznach et al., 2014), for GLM it is not possible to estimate time travel, since inflows are directly placed at defined layers. Studies assessing the inflow pathways through modeling demonstrated a good agreement between simulations and observations, for example Marti et al. (2011) and (Zamani et al., 2020) with 3D models and Jeznach et al. (2014) with CE-QUAL-W2.

Despite similar seasonality of stratification and mixing predicted by the three models, the tracer analysis demonstrated how the

spatial simplification affects the transport of substances. This was clear especially for the 1D model that differed considerably from the 2D and 3D models. CE-QUAL-W2 and Delft3D presented similar seasonal patterns for density currents, which influenced the stratification in the reservoir mainly by underflows that added a layer of colder water at the bottom and were strongly present in the simulations during winter. Ishikawa et al. (2021a) analyzed the distribution of density currents, categorized as under, inter and overflows, based on the comparison of the measured temperature between the forebay region and the main reservoir (only available

for the first half of the total period) with the temperature profile at the intake. Overflows were assigned when the forebay temperature was larger than the surface of the measured profile, underflows when lower than the bottom, and otherwise they were interflows. A similar classification of density currents was made using the location of the maximum tracer concentration of the models (Fig. SI 2). Processes such as entrainment, mixing, diffusion and dilution of inflow are neglected in this approach, therefore overflows were frequent in March and April 2018, while of minor importance in the simulations. However the underflow season

that was expected in the prior analysis was at some extension reproduced by CE-QUAL-W2 and Delft3D. It was mainly present during the winter, with shifts of one month among analyses made based on observations and simulations, matching with the period where the process is relevant for stratification.

For the second half of the simulated annual cycle, the flow paths of 2D and 3D models differed the most in respect to the occurrence of overflows, which is explained by the location of the density currents in relation to the UML depth (Figure 12 and Fig. SI 2). All

models had poor results for UML depth estimated from measurements. For this reason, the 2D model presented a smoothed error along the depth (Figure 5), while the 1D and 3D models simulated consistently shallower UML depths causing the peak in the error and correlation profiles. The prediction of thickness of the mixed layer and the slope of the thermoclines are generally challenging for models, as reported for other model applications (Perroud et al., 2009; Huang et al., 2010).

### 7.4 Flow velocities and vertical mixing

Simulation of flow velocities showed less agreement with measurements than temperature, although errors were in the same range of other works with Delft3D (Chanudet et al., 2012; Dissanayake et al., 2019). The magnitudes of simulated longitudinal flow velocities were generally lower than observations, but Delft3D was capable to reproduce the overall characteristic of larger magnitudes of longitudinal flow velocities during stratification, and larger relative occurrence of currents ($> 90$[th] percentile), while flow velocities simulated by CE-QUAL-W2 showed no agreement in magnitude and dynamics with observations. For a fair

comparison with the 2D model, laterally averaged flow velocity observations would be required, which are not accessible from longer-term observations. However the transversal flow velocity component of observations and simulations of Delft3D were disregarded in the comparison ($0.96 \pm 0.98$ cm s$^{-1}$ and $0.54 \pm 0.64$ cm s$^{-1}$, respectively). The ratio of mean transversal and





longitudinal velocity components are 0.7 for measurements and 0.5 for Delft3D, which indicates that potential transversal flow processes are not resolved by CE-QUAL-W2.

The poor results regarding magnitude and direction of flow velocities in both models can be associated to the fact that flow velocities in Passaúna Reservoir were generally small, internal seiches were not observed and circulation patterns are absent. In studies where those properties are relevant, better agreement between observations and 3D simulations were reported (Huang et al., 2010; Chanudet et al., 2012; Dissanayake et al., 2019), and a simulation in Lake Erie with CE-QUAL-W2 reproduced the oscillation frequency of basin scale seiches, but not their amplitudes (Boegman et al., 2001). Further, the direction of flow velocities

are highly affected by the inaccuracy of wind direction measurements, and weak wind intensities are expected to have lower correlation with flow velocities, since it potentially changes direction more frequently (Dissanayake et al., 2019), which is the case in Passaúna. Moreover, the correlation between observed and simulated flow velocities increased towards the bottom, probably because the bottom is a better represented boundary for the process, whereas it is the opposite is the case for temperature.

The turbulent closure models are relevant for the vertical mixing, thus the different approaches implemented in the models can be

considered to contribute to the variation among simulated UML thicknesses. GLM is the one with the thinner UML and poorest correlation, its simplification in dimensionality and the structure of the model in mixed layers can be the cause for lower mixing. CE-QUAL-W2 neglects transversal flows, which may contribute to the shear profile and cause errors in turbulence production. Lastly, Delft3D resolved all three dimensions, but dissipation rates simulated by the model differ from those estimated from measurements of flow velocities. The log-averaged profiles of dissipation rates computed by the 3D model follow an expected

profile with a turbulent surface boundary layer, a low energetic interior and increasing dissipation rates towards the bottom (Wüest and Lorke, 2003). Only few studies in the literature reported comparisons of measured and simulated dissipation rates. In general they presented good agreement, but all of them were performed in high energetic environments such as ocean regions with the presence of breaking waves, near to the surface, and large lakes (Stips et al., 2005; Jones and Monismith, 2008; Paskyabi and Fer, 2014; Moghimi et al., 2016). In spite of all uncertainties regarding the estimated dissipation rates, in our study the results from the

3D simulations revealed that the model has limitations in reproducing all processes contributing to energy dissipation in a medium sized reservoir.

**8 Conclusions**

Three commonly used hydrodynamic models with different dimensionality were applied to a subtropical reservoir using identical boundary conditions and simulation results were compared to measurements covering a complete annual cycle. All models were

capable of providing valuable information about the water balance and reproduced the overall pattern of seasonal thermal stratification and mixing. Flow velocities, only available from the 2D and 3D models, were more challenging to reproduce, particularly because of low flow velocities and lack of large-scale circulation pattern in the reservoir. In terms of MAE for water level, temperatures and flow velocities, the three models had a maximum variation among each other of 30%, but the time required to run the simulations increased by nearly five orders of magnitude, from 5 s with GLM, to 3.7 min with CE-QUAL-W2 and 3.5

days with Delft3D (Table 1). Passaúna is a medium-sized reservoir, therefore for larger systems computational time can turn into a more constraining factor.

Nevertheless, each model has its advantages and limitations and their application should be chosen in accordance with the parameters to investigate



1D:

- Water balance / water level, fundamental for the management of reservoirs.

- Seasonal operations that depend on stratification, e.g. selecting intake and outflow depths that will have better results depending on the mixing condition (Weber et al., 2017).

- The good trade-off between computational costs and provided accuracy in simulating seasonal thermal stratification and vertical mixing is attractive, and 1D models have been increasingly employed in larger-scale studies including a large

number of water bodies (e.g. Read et al., 2014; Woolway and Merchant, 2019)

2D:

- Assist in actions and time response regarding substance transport, e.g. in case of contamination of the principal inflow (Jeznach et al., 2014). However specific determination of layers containing the density currents are uncertain, it will depend on initial mixing of inflow and depths of UML and thermocline, which none of the models could reproduce with

precision, although the 3D had better correlation.

- Seasonal pattern of the density currents.

3D:

- Field of flow velocities. Although Passaúna Reservoir had low kinetic energy, the 3D model presented positive correlation with measurements. In addition, wind speeds were low and were measured a few km apart from the monitored site, where

it could have different direction and reduce the agreement between observation and simulation. Flow velocities can be important for processes that depend on circulation patterns, for example the transport of nutrients that are related to algae blooms (León et al., 2005; Chung et al., 2014).

Challenges faced by all models were the water balance and the UML thickness. The first rather because of the poor monitoring, thus it is of paramount importance to have good measurements of the volumetric discharge of inflows, and outflows, including

engineering structures, such as spillways. Otherwise it is difficult to identify sources of errors related to the models itself. The thickness of the mixed layer can have large effects subsequent simulations of water quality. The categorization of density currents in over, inter and underflow depend on that and will have a direct impact on the fate of nutrients and organic matter inside the reservoir (Rueda et al., 2007). Similarly, the dynamics and vertical distribution of dissipation rates of turbulent kinetic energy could not be reproduced. This quantity can be relevant not only for hydrodynamic applications, but also for the prediction of air-

water gas exchange (Katul and Liu, 2017), sediment-water fluxes (Lorke and Peeters, 2006; Grant and Marusic, 2011) or the development of algal blooms (Aparicio Medrano et al., 2013). By taking these general and model-specific limitations into consideration, models are valuable tools not only for managing water resources, but also for scientific applications (e.g. Sabrekov et al., 2017; Mi et al., 2020; de Carvalho Bueno et al., 2021).

*Data availability:* If accepted, the data will be uploaded to a repository.

*Author contribution:* MI: conceptualization, application of Delft3D, formal analysis, and investigation. WG: set up of Delft3D, adjustment of grids, bathymetry, and analysis of water balance. OG: first set up of Delft3D and application of CE-QUAL-W2. GS and J.AR application of GLM. TB: supervision of GS and OG master thesis, advisor in all models and concepts. MM: co-



supervision of GS, and advisor for the ms concept. AL: supervision of MI PhD thesis, advisor for the ms concept. Writing – original draft preparation was performed by MI, WG, J.AR, GS, and OG. Writing – review & editing: AL, TB, and MM.

*Competing interests:* The authors declare that they have no conflict of interest.

*Acknowledgements:* All measurements and analysis presented in this paper were part of the MuDak-WRM project: https://www.mudak-wrm.kit.edu/ (Fuchs et al., 2019), funded by the German Federal Ministry of Education and Research (BMBF), under the grant number 02WGR1431 B. Project partners provided data to support this specific study, such as the bathymetry and the inflows discharges and temperatures. We also thank SANEPAR and the Postgraduate Program in Water
Resources and Environmental Engineering from the Federal University of Paraná, for collaborating during the field work and providing data. Tobias Bleninger acknowledges the productivity stipend from the National Council for Scientific and Technological Development – CNPq, grant no. 312211/2020-1, call no. 09/2020.

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
