# Peer review of "Effects of dimensionality on the performance of hydrodynamic models for stratified lakes and reservoirs"

_Geoscientific Model Development, 2021_

## Author Comment (AC4)

**In the following response, the original review is shown in black and our response in orange.**

**Comment on gmd-2021-250**
Victor Stepanenko (Referee)

Referee comment on "Effects of dimensionality on the performance of hydrodynamic models" by Mayra Ishikawa et al., Geosci. Model Dev. Discuss., https://doi.org/10.5194/gmd-2021-250-RC2, 2021

The paper brings valuable information for the limnological community on the relative performance of models of different spatial dimensionality applied to an artificial reservoir with significant longitudinal variation of water properties. 1D, 2D and 3D models are compared to measurements carried out at Brazilian drinking reservoir, in terms of water level, temperature, velocity, turbulence and passive tracer. Results elucidate capabilities and limitations of models used, which allowed authors to formulate recommendations on their further applications. I suggest to publish the paper after minor revisions.

We thank Victor Stepanenko for his valuable comments to improve the study.

My two major concerns on the paper are:

What was the model calibration procedure? Please clearly describe for each model. Usually, in intercomparison studies some parameters are set the same for all models, the others are allowed to be calibrated individually. Please provide reasoning on the choice of such parameter groups in your case.

A better description of the calibration procedure will be provided as a subsection of section 4. We indeed predefined some parameters to be equal in all models, but later we noticed that the model performance can be improved by adjusting their values as part of a model calibration. By now we can say that the main parameters were the bulk exchange coefficients for heat and momentum (wind drag coefficient), which are the typical calibration parameters of models.

1D models by construction simulate the horizontally averaged fields, e.g. temperature; this means it is strictly speaking incorrect to state that 1D model simulates the temperature of point observations better or worse, than 2D/3D model, because there is a possibility in a latter case to take temperature from a cell nearest to observation location; this is what should be mentioned in discussion on results of the model intercomparison for intake region; my suggestion is also to add 1D model

comparison to horizontally averaged data from reservoir-wide surveys which you use in Section 6.2.2

We agree with the comment and will follow your suggestion. We will add a comparison of horizontally averaged temperature from the 2D and 3D models, and estimate the error with the measured temperature profile.

Specific comments:

The title: hydrodynamic models of what? (reservoir?)
We agree that the type of system needs be better specified, as the scope of the journal is very broad. We will change it to: Effects of dimensionality on the performance of hydrodynamic models for stratified lakes and reservoirs.

Line 29: please change "identical"
It will be changed to: "While the mechanistic description of underlying physical processes are similar in all models …"

Line 125: I suggest to change: lateral -> transversal
The wording will be changed accordingly.

Line 130: explain the choice of turbulent scheme in this study
Information will be added. We simply chose the most common model (k-ε), which was also available for the 3D model.

Line 135: shallow water equations are 2D in space; this is not the same as 3D dynamics with hydrostatic approximation
The shallow water equations used in Delf3D are 3D in space. It is assumed that the horizontal length scales are much larger than the vertical ones. But the third dimension (z) is considered and the calculated parameters change over all 3 dimensions (i.e. 3D-application).
The momentum equation in z-direction is simplified by considering negligible vertical acceleration, which leads to the hydrostatic equation for pressure, i.e. hydrostatic pressure assumption.  Then the vertical velocity is calculated via the continuity equation for the case of 3D models.
We will rewrite to: "3D Reynolds-Averaged Navier-Stokes equations with the hydrostatic approximation for the vertical direction".

Line 138: I suggest to change: resolved -> parameterized
The wording will be changed accordingly.

Line 169: As one can judge from this section, downwelling longwave radiation was not measured and used to force models, rather empirical formulae applied; this might be one of error sources, please indicate in discussion

We will follow the suggestion and mention the lack of longwave radiation measurements as one source of uncertainty in the discussion. However, detailed analysis of the different formulations used by the models to estimate the radiation balance is certainly beyond the scope of the present manuscript.

Table 1: I suggest to change: wind coefficient -> drag coefficient

The wording will be changed accordingly.

Table 1: the light extinction coefficient was put different in models; have you had any measured transparency properties like Secchi disk?

This was one of the parameters that initially was going to be the same in all models and later it was decided to be used as a calibration coefficient. Secchi disk depths were measured over the campaigns, and its average along the longitudinal and over time was 2 m, which by coincidence is the default in Delft3D, and leads to a light extinction coefficient of 0.85 m$^{-1}$. This information will be added to the calibration description.

Table 1: "branches width" or "segment width" in 2-d raw?

A branch is a collection of segments, in our case we had 2 branches. The first is the main axis of the reservoir, and the second is the left side arm. So it is the segment width. An indication of the branches will be added in Figure 3 to make it clear.

Table 1: what is 0.85 m in raw 3? grid spacing?

The thickness of the vertical cell.
We will change the wording in the first column to make it clear.

Table 1: specify compiler in computational time section

We will add the information that all source codes were written in FORTRAN.

Table 1: longwave radiation schemes are different, what are the implications?

See our response to your former comment. We will follow the suggestion and mention the lack of longwave radiation measurements as one source of uncertainty in the discussion. However, detailed analysis of the different formulations used by the models to estimate the radiation balance is certainly beyond the scope of the present manuscript.

Table 1: "Kinematic viscosity of water" -> did you mean molecular viscosity?
Yes. We will make this clearer by rephrasing to "molecular kinematic viscosity"

Table 1: Lines "Vertical eddy viscosity" and "Vertical eddy diffusivity" should contain coefficients not simulated by "Turbulence closure model", namely, background values; this is not clear by formulation "Computed", etc.
We will remove the word computed and provide only the background values of viscosity and diffusivity.

Table 1: please replace "Computed ..." by concrete information on computation scheme
We believe that a description of the computational scheme (especially for diffusivity) would become too complex for Table 1. We will modify the table to list the background values of eddy viscosity and diffusivity, respectively.

Heat exchange with sediments neglected, what are implications, esp. for shallow zones?
We briefly mention this in the discussion at Line 517, where it says: "Regarding the heat exchange with sediment, Stepanenko et al. (2013) showed that it did not have significant influence on simulations of bottom water temperature of a shallow lake for a comparable temperature range as observed in Passaúna Reservoir."

Line 223: water level is not a boundary condition (understanding boundary conditions in mathematical sense as additional constrains at boundaries for partial differential equations)
The water level at the spillway was used as a boundary condition as it represents an open boundary in the 3D model. This information will be added to the manuscript.

Line 325: do evaporation differences explain level discrepancy between models?
This can be one of the explanations, but since we do not have good measurements of evaporation it is not possible to affirm which model was the best. This was discussed at Line 500.

Line 364: p-value is given for which hypothesis? Please clearly explain so that the reader understands the hypothesis being tested every time you mention p-value
The information will be added accordingly.

Fig. 6 b,c,d: better to add regression line
Regression lines will be added.

Lines 466-468: I can't follow this sentence

We will rephrase the sentence: "The GLM model was set up with a maximum water depth of 17 m, while at the point of analysis water depth was ~12 m. For this reason, we present model outputs up to a maximum depth of 12.5 m. If the maximum of the tracer concentrations was below this depth, the inflow regime was categorized as underflow. For CE-QUAL-W2 and Delft3D the closest cell to the station was selected, which represents the actual water depth at the monitoring site."

Line 469: what is interflow, underflow, overflow?

A small description of them will be added at the beginning of the section. They are a classification of the flow path of the inflowing waters within the reservoir according to their location over the depth. Overflows have a path along the surface, underflows along the bottom and interflow in intermediate depths.

Lines 483-485: better to put in beginning of section as definitions of terms used

Agreed, as mention in the previous answer.

---

## Author Comment (AC5)

**In the following response, the original review is shown in black and our response in orange.**

**Comment on gmd-2021-250**
Laura M. V. Soares (Referee)

Referee comment on "Effects of dimensionality on the performance of hydrodynamic models" by Mayra Ishikawa et al., Geosci. Model Dev. Discuss., https://doi.org/10.5194/gmd-2021-250-RC1, 2021

General comments:

The authors applied 1D, 2D, and 3D numerical models running with identical initial and boundary conditions to simulate hydrodynamic processes in a medium-sized drinking water reservoir. The results of the models supported a further understanding of how dimensionality affects model performance and the authors highlighted which dimensions are better suited for representing different hydrodynamic processes.

I think the authors have created a very interesting manuscript, and that the findings presented here have the potential to make a good contribution to the literature. It should be well received by model users and by a broad audience of the Geoscientific Model Development journal. The manuscript is well-structured, includes valuable and useful figures, and is based on relevant and recent literature. The models are well-described, the authors indicated which version was applied, and the results are sufficient to support the interpretations and conclusions. I recommend the acceptance of the work for publication after a number of comments have been addressed. I have few scientific questions/issues about the methodological approach and additional pieces of information that I believe should be included in the manuscript, and a list of purely technical corrections. I have outlined my comments below as either specific comments (relating to the methods/findings) or technical corrections (relating to word choice or content organization).

We thank Laura M. V. Soares for her valuable comments to improve the study.

Specific comments:

- Introduction Page 2, Lines 49–50: I suggest the authors expand a bit further that the selection of a model's dimension is ultimately defined by the research question and must be bound by justifiable simplifications to balance the variable of interest, the ecosystem, and knowledge, not making the model more complex than the data set

can support. This would help the reader understand the recommendations listed by the authors in the conclusions regarding the suitability of 1D, 2D, and 3D models for representing different hydrodynamic variables.

We agree that the selection of the model should be defined according to the research question and environment. We will add an explanation to the revised introduction.

- P 6, L 178: How could the modeled discharge of Passaúna River be calibrated? Is there a discharge gauge station on the Passaúna River? If so, please indicate in Fig. 1.

The hydrological model and its calibration was done, and is described in a companion publication using data from a gauging station upstream of the reservoir (Ishikawa et al., 2021b). We will add this information to the revised version of the manuscript and include the location of the gauging station to Fig. 1.

- P 7, L 213: I am wondering if your approach of manual calibration is somehow biased as the models require quite different efforts to this procedure, for instance, calibration of the 1D model demands much lower effort than for the 3D model. My reasoning is that if an automatic calibration was performed applying the same calibration range and the same number of iterations in each model, the results would be better comparable. Perhaps, a brief discussion should be included about how the manual calibration could affect the results.

As both reviewers raised concerns about the calibration process, we added a more detailed description to the revised manuscript. Although automatic calibration is available for GLM we did not use it, therefore its calibration processes was similar to that of the 2 and 3 D models. Nevertheless, it is worth to mention that 1D models are more easy to calibrate and can complement models of higher dimensionality. We discuss this issue in line 530 ff. of the manuscript.

- Table 1: Why did the authors adopt different time steps for each model? Wouldn't it be possible to adopt the same time step for the 3 models? Do you envisage how this might influence model performance?

In the 2D and 3D models, the numerical time steps were similar (1 and 12 seconds). In the 3D model, the time step was selected based on the Courant number in order to meet numerical stability. Shorter time steps increase computational costs, but do not change the results, once numerical stability is achieved. In the 1D model, we used the recommended time step of 1 hour, which is the default value of GLM and common practice, as observed in several other works using GLM (Farrell et al., 2020; Gal et al., 2020; Ward et al., 2020; Ladwig et al., 2021).

Due to the valuable reviewer comments we tested different time steps (1, 12, 30, 60, 1800, and 86400 seconds) with GLM. To our surprise, even not presenting numerical stability issues the results were quite different using different time steps.

According to Hipsey et al. (2019) at section 2.1: "*Surface mass fluxes operate on a sub-daily time step, dt, by impacting the surface layer thickness (described in Sect. 2.2), whereby the dynamics of inflows and outflows modify the overall lake water balance and layer structure on a daily time step, $dt_d$, by adding, merging, or removing layers (described in Sect. 2.7).*" Furthermore, on p. 487 some damping mechanisms are applied, depending on time-steps.

Therefore, we concluded that changing the time steps in GLM changed the number of layers and affected the mixing characteristics. Thus, time step changes become a calibration parameter. According to our tests the time step which best fits to the measurements was 1800 seconds (cRMSE = 0.83°C), and simulations with $\Delta t = 3600$ seconds was in the same order of magnitude (cRMSE = 0.84 °C), see figure below. Since the analysis in the manuscript was made with $\Delta t = 3600$ seconds and error difference with the best simulation is minor, we believe that it is not necessary to change time step for a revised version.

[Figure]

*Figure 1: RMSE vs Δt (numerical time step) of simulated temperatures by GLM*

The different time steps should not be a problem for the model comparison, once the models are stable and calibrated, for example the model intercomparison presented by Stepanenko et al. (2014) used time steps varying from 30 to 3600 seconds. In addition, if the input data does not have a better temporal resolution it is

not reasonable to decrease the time step, once the model is already stable and smaller time steps increase the computational time (GLM with dt = 1 sec took 55 minutes to run the simulation) and models usually interpolate linearly between the given measured time steps.

We can add this information over the calibration description that we already planned to include in the revised version. The figure can be presented at supplementary information.

- Section 5: The key point of the manuscript is the performance assessment of the hydrodynamic models based on statistical metrics calculated for the variables of interest. However, the description of the indices for comparison is somehow incomplete and must be clearly outlined. For example, in section 5 – indices for comparison, the authors present: stratification duration based on the ST, UML, temperature, and flow velocities. But the authors also analyzed other variables beyond the above-mentioned: water level, spillway discharge, evaporation rate, the formation of currents, and substance transport. Some of those variables appear in the Results section for the first time, but they should be introduced in section 5. Also, in section 5.1 – Statistics the authors present the following metrics: standard deviation, r, cRMSE, and MAE. However, not all statistics are present for the variables of interest. For example, I missed the standard deviation, r, and cRMSE for water level. In addition, other statistics were applied by the authors (coefficient of determination, percentiles, and percentage difference) and they should be stated in section 5.1. Hence, I recommend the authors describe all statistics and variables of interest in section 5 aiming at a better structure of the methods and thus the reader can better follow the results. Perhaps, adding a table synthesizing all statistic metrics for each hydrodynamic variable would help to visualize the results.

The information about water level, spillway discharge and evaporation rate were not presented at section 5 because they are direct results (a simple time series) and not derived quantities (like stratification and currents). In the revised manuscript, we will mention all variables of interest in section 5 and add more specific information on the formation of currents. In addition, we will provide a complete description of the statistical indices that are used for comparison.

A table presenting all variables with corresponding statistical metrics will be added to supplementary information.

- P 21, topic 6.3.3. In section 4, the authors explain that the tracers are implemented starting from 1 Aug 2018. Could you please explain the presence of the tracers in the intake region since the beginning of the simulation period (March 2018)?

It started on 1 Aug 2017, thank you for noticing the typo. It will be corrected.

Technical corrections:

- P 1, L 35: "land use" instead of "land usage"?
The wording will be changed accordingly.

- P 2, L 51: "as well as to the assessment" instead of "as well to assessment"?
The wording will be changed accordingly.

- P 2, L 67: This sentence is not a conclusion from the work of Polli and Blenninger, 2019, neither from Soares et al., 2019. I suggest removing these citations from here.
It will be removed.

- P 2, L 69-70: What do you mean by "good results"?
It will be changed to: "better agreement with measurements".

- P 3, L 84-85: The sentence is disconnected from the idea of this paragraph and its content is more close to the idea of lines 49-50. I suggest the authors move these lines to be closer to line 50.
It will be changed accordingly.

- P 5, L 153: The format of the reference here is not correct. Could you correct it, please?
It will be corrected.

- Fig. 1: what is PPA?
The name of the monitoring point. A note will be added in the legend.

- Fig. 2: I missed the time-series of rainfall. Could you provide it, please?
Yes, this information can be provided.

- P 6, L 179: "beseflow" instead of "baseline"?
The wording will be changed accordingly.

- Table 1: What does the * mean in the second line of GLM column?
An indication that the thickness is not fixed. A note will be added at the end of the Table.

- P 9, L 226: The format of the reference here is not correct. Could you correct it, please?
It will be corrected.

- P 11, L 260: The format of equation 2 is not correct. Could you revise it, please?
It will be corrected.

- P 11, L 276: "only for the period" instead of "only the period".
The wording will be changed accordingly.

- P 12, L 294: The format of the reference here is not correct. Could you correct it, please?
It will be corrected.

- P 13, L 334-335: Could you rephrase that line ("Persistent thermal stratification developed in spring, and retained over summer"), please?
It can be rewritten to: "Thermal stratification developed in spring and persisted throughout the summer".

- Figure SI 1a: unit "m.a.s.l." instead of "m".
The wording will be changed accordingly.

- Fig. 4: Why simulation results of GLM is at 0.5 m depth rather than at 1 m depth to be at the same depth of measurements? It would provide a better comparison between (a) and (b). Also, by a visual inspection, it seems that temperature simulated by GLM is higher than the measured. If the authors use the simulated temperature by GLM at 1 m depth, the same depth of measurements, the contour plots would be better comparable.
Results from GLM were linearly interpolated to a fixed Δz = 0.5 m because the thickness of the cells change over time. This procedure should not have a great impact on the results. Indeed GLM simulated larger surface temperatures, the statement in line 355 was based on calculations of temperatures at the same depth of the measurements.

- Fig. 4 caption: "intake" instead of "Intake".
The wording will be changed accordingly.

- P 15, L 335: "0.5 °C" instead of "0.5°C".
The wording will be changed accordingly.

- P 15, L 359: "Schmidt stability" instead of "Schmidt number".
The wording will be changed accordingly.

- P 15, L 363: The correlation coefficient (r) rather than the coefficient of determination (R2) is a better metric to provide a measure of the correlation between simulated and observed variables. The same is valid for P 15, L 366.
We will provide the correlation coefficient.

- Please review the citation of figures along the text. For instance, Fig SI 3 is cited in the text before Fig SI 2; Fig. 10 is cited in the text before Fig. 9; and Fig. 7 is not cited in the text.
It will be revised.

- P 19, L 419: "deviation" instead of "deviations".
The wording will be changed accordingly.

- P 20, L 443: "was" instead of "were".
The wording will be changed accordingly.

- P 23, L 491-492: The sentence presents results and should be placed on Results section rather than in Discussion section.
The results are just presented in another form, because they were estimated through water level presented in results section. We thought it could be good to present the result again so the reader does not need to go back in the paper and have a new perspective on it while reading discussion.

- P 23, L 492: "water level is similar" instead of "water level similar".
The wording will be changed accordingly.

- P 24, L 529: "strength of vertical" instead of "strength if vertical".
The wording will be changed accordingly.

- P 25, L 562: The format of the reference here (Zamani et al., 2020) is not correct. Could you correct it, please?
It will be corrected.

- P 26, L 628: The sentence lacks punctuation. Could you correct it, please?
It will be corrected.

- P 27, L 651: "large effects on subsequent simulations" instead of "large effects subsequent simulations".
The wording will be changed accordingly.

- Could you please provide the DOI for the following references, please: Chung et al. 2014, Dai et al., 2013; Kobler et al., 2018; Lorke and Peeters, 2006.
They will be provided.

References:

Farrell, K. J., Ward, N. K., Krinos, A. I., Hanson, P. C., Daneshmand, V., Figueiredo, R. J., and Carey, C. C.: Ecosystem-scale nutrient cycling responses to increasing air temperatures vary with lake trophic state, Ecological Modelling, 430, 109134, https://doi.org/10.1016/j.ecolmodel.2020.109134, 2020.

Gal, G., Yael, G., Noam, S., Moshe, E., and Schlabing, D.: Ensemble Modeling of the Impact of Climate Warming and Increased Frequency of Extreme Climatic Events on the Thermal Characteristics of a Sub-Tropical Lake, Water, 12, 1982, 2020.

Hipsey, M. R., Bruce, L. C., Boon, C., Busch, B., Carey, C. C., Hamilton, D. P., Hanson, P. C., Read, J. S., de Sousa, E., Weber, M., and Winslow, L. A.: A General Lake Model (GLM 3.0) for linking with high-frequency sensor data from the Global Lake Ecological Observatory Network (GLEON), Geoscientific Model Development, 12, 473-523, https://doi.org/10.5194/gmd-12-473-2019, 2019.

Ladwig, R., Hanson, P. C., Dugan, H. A., Carey, C. C., Zhang, Y., Shu, L., Duffy, C. J., and Cobourn, K. M.: Lake thermal structure drives interannual variability in summer anoxia dynamics in a eutrophic lake over 37 years, Hydrol. Earth Syst. Sci., 25, 1009-1032, 10.5194/hess-25-1009-2021, 2021.

Stepanenko, V., Jöhnk, K. D., Machulskaya, E., Perroud, M., Subin, Z., Nordbo, A., Mammarella, I., and Mironov, D.: Simulation of surface energy fluxes and stratification of a small boreal lake by a set of one-dimensional models, Tellus A: Dynamic Meteorology and Oceanography, 66, 21389, https://doi.org/10.3402/tellusa.v66.21389, 2014.

Ward, N. K., Steele, B. G., Weathers, K. C., Cottingham, K. L., Ewing, H. A., Hanson, P. C., and Carey, C. C.: Differential Responses of Maximum Versus Median Chlorophyll-a to Air Temperature and Nutrient Loads in an Oligotrophic Lake Over 31 Years, Water Resources Research, 56, e2020WR027296, https://doi.org/10.1029/2020WR027296, 2020.

---

## Author Response (AR1)

Dear editorial team,

we are submiting the revised version of the manuscript gmd-2021-250 entitled "Effects of dimensionality on the performance of hydrodynamic models".
Please note that the detailed point-by-point response to all referee comments was provided over discussion (https://gmd.copernicus.org/preprints/gmd-2021-250#AC5 and https://gmd.copernicus.org/preprints/gmd-2021-250#AC4). Therefore, as agreed with the topical editor, at the moment only the revised version with the marked changes is being provided, which was done according to the given responses.

Kind regards,
Mayra Ishikawa

---

## Author Response (AR2)

Dear editorial team,

we are submiting the most update version of the manuscript gmd-2021-250.
Attending to the topical editor comment, the only change is in Table 1 and its caption. Which now explicit state that the calibrated parameters are marked with a sign within the table.

Kind regards,
Mayra Ishikawa